# Application of AHP-ICM and AHP-EWM in Collapse Disaster Risk Mapping in Huinan County

Zengkang Lu [1], Chenglong Yu [1,*], Huanan Liu [2], Jiquan Zhang [3], Yichen Zhang [1], Jie Wang [1] and Yanan Chen [1]

1 College of Jilin Emergency Management, Changchun Institute of Technology, Changchun 130012, China; 20221311204@stu.ccit.edu.cn (Z.L.); zhangyc@ccit.edu.cn (Y.Z.); 20221311210@stu.ccit.edu.cn (J.W.); chenyn061@nenu.edu.cn (Y.C.)
2 School of Prospecting and Surveying, Changchun Institute of Technology, Changchun 130021, China; liuhuanan@ccit.edu.cn
3 School of Environment, Northeast Normal University, Changchun 130024, China; zhangjq022@nenu.edu.cn
* Correspondence: yuchenglong@ccit.edu.cn; Tel.: +86-1350-088-8781

**Abstract:** Collapses are one of the most common geological disasters in mountainous areas, which easily damage buildings and infrastructures and bring huge property losses to people's production and life. This paper uses Huinan County as the study area, and with the help of a geographic information system (GIS) based on the formation principle of natural disaster risk, the information content method (ICM), the analytical hierarchy process (AHP), and the analytical hierarchy process–information content method (AHP-ICM) model are applied to hazard mapping, and the analytical hierarchy process-entropy weight method (AHP-EWM) model is applied to exposure, vulnerability and emergency responses, and recovery capability mapping. A risk mapping model for collapse disasters was also constructed using these four elements. Firstly, an inventory map of 52 landslides was compiled using remote sensing interpretation, field verification, and comprehensive previous survey data. Then, the study area mapping units were delineated using the curvature watershed method in the slope unit, and 21 indicators were used to draw the collapse disaster risk zoning map by considering the four elements of geological disaster risk. The prediction accuracy of the three hazard mapping models was verified using the receiver operating characteristic (ROC) curve, and the area under the curve (AUC) results of the AHP, ICM, and AHP-ICM models were 80%, 85.7%, and 87.4%, respectively. After a comprehensive comparison, the AHP-ICM model is the best of the three models in terms of collapse hazard mapping, and it was applied to collapse risk mapping with the AHP-EWM model to produce a reasonable and reliable collapse risk zoning map, which provides a basis for collapse management and decision making.

**Keywords:** risk mapping; GIS; analytical hierarchy process; information content method; spatial analysis

## 1. Introduction

Geological hazards such as collapses and landslides often occur in mountainous areas and can pose a major threat to the safety of life, property, and transport infrastructure in areas of intense human activity [1,2]. A collapse is a phenomenon in which a mountain, earth or rock deposit, or building loses its stable balance and collapses partially or completely due to geological action or external forces, mostly among steep-angled mountains [3], and may be caused by factors such as heavy rainfall [4], seismic activity [5], and human engineering activities [6] (mining, slope excavation, etc.). To better identify areas where collapse disasters may occur and to take effective prevention and control measures, risk mapping of collapse disasters is essential [7–9].

In recent years, driven by the development of mountainous cities, many road construction projects require collapse hazard and risk evaluation to protect people's lives and properties [10–12]. According to statistics, in 2017 alone, more than 2800 geological disasters occurred in China, resulting in hundreds of deaths and economic losses of more than CNY

1 billion. Among them, repeated collapse disasters occurred in some mountainous areas, destroying a large amount of infrastructure and farmland and causing a large number of rural people to lose their production and livelihood. Although the size of rock masses is relatively small, these rock falls can instantly cause huge losses and hazards due to their suddenness and the huge impact they contain, especially along roads [13].

At present, great progress has been made in the work of collapse disaster risk evaluation, which mainly consists of three major categories: qualitative analysis, quantitative analysis, and a combination of both [14–19]. The analytical hierarchy process (AHP) is a more widely used method for qualitative analysis of collapse disaster risk evaluation [18,20,21], which is a multi-objective decision-making method with a high degree of subjectivity for determining the degree of contribution of each collapse indicator to collapse risk. Quantitative analysis is a method of analysing problems using mathematical models, which takes objective data as input and applies tools such as mathematical and statistical analysis to calculate the results and analyse them, such as the information content method (ICM) [22], frequency ratio method (FRM) [17,23], entropy weight method (EWM) [24], and logistic regression (LR) [25,26]. Using only one method is too subjective or relies too much on mathematical models and yields somewhat one-sided weight values. However, combining the two types of weight-defining methods can make up for the single deficiency and achieve the integration of subjective and objective and qualitative and quantitative. In this study, AHP is combined with ICM and EWM, respectively, based on the advantages of different methods and ArcGIS 10.8.1 to investigate the riskiness of collapse disasters in Huinan County.

In collapse risk mapping, the selection of a suitable mapping unit is a key step. At present, more and more scholars choose slope units based on the characteristics of collapses [22,27], while the hydrological analysis method is more cumbersome in GIS extraction, and the method cannot identify the horizontal plane correctly. To give a more complete picture of the slope where the collapse site is located and thus be able to carry out a more practical and reasonable division and improve the model prediction accuracy, the curvature watershed method [25] was chosen for the unit division of the mapping unit in this study.

The purpose of this study is to provide a basis for the establishment of the riskiness model of collapse in Huinan County, Tonghua City, Jilin Province. Based on the 1:50,000 geological hazard survey in Huinan County, a total of 52 collapse disaster sites in the study area were counted. The curvature watershed method was used to divide the entire study area into units and then apply the formation principle of natural disaster risk [28]. Evaluation indexes were selected from four standard layers: disaster hazard, exposure, vulnerability, and emergency response and recovery capability of the disaster bearers, and correlation analysis was performed on the selected indicators. Then, the AHP, ICM, and integrated AHP and ICM methods were used to establish the hazard mapping models, and the AHP-EWM method was used to establish the exposure, vulnerability, and emergency response and recovery capability mapping models. Finally, the predictive ability of the three hazard mapping models was verified by using the receiver operating characteristic (ROC) curve, and the optimal model was selected and plotted with the AHP-EWM model. The risk map of Huinan County was created, and risk mapping of collapse disasters in Huinan County was performed. The results of the study will provide guidance for risk management and disaster prevention and mitigation in Huinan County and provide a reference for policymakers.

## 2. Data and Methodology

### 2.1. Overview of the Study Area

The research area is in Huinan County, Tonghua City, Jilin Province (Figure 1), with a total area of about 2277 km$^2$ and geographical coordinates at 125°58′–126°45′ E and 42°16′–42°54′ N. The terrain is characterised by high southeast and low northwest points, with the highest point at 1222 m and the lowest point at 187 m above sea level. The

climate of the study area is a north temperate continental monsoon, with an average annual precipitation of 696.3 mm according to meteorological statistics from 1998 to 2014, with precipitation mainly concentrated in the period of June to August. Field investigations show that the landslides are mainly located along the Huinan–Jingyu highway and railway, as well as in the mining areas of Chaoyang Town, Huinan Town, Fumin Town, etc. The types of slopes forming the landslides are mainly artificial rocky slopes, followed by natural rocky slopes. The geotechnical types are more complex, mainly basalt (β), granite (γ), hard clastic rock (Q + Z), softer clastic rock (K + J), limestone (∈ + O + Z), metamorphic rock (As) and clay and gravel double-layered soil (Q). Possible destabilising factors are rainfall, frost heave, and human damage, with collapse mostly occurring during the rainfall process.

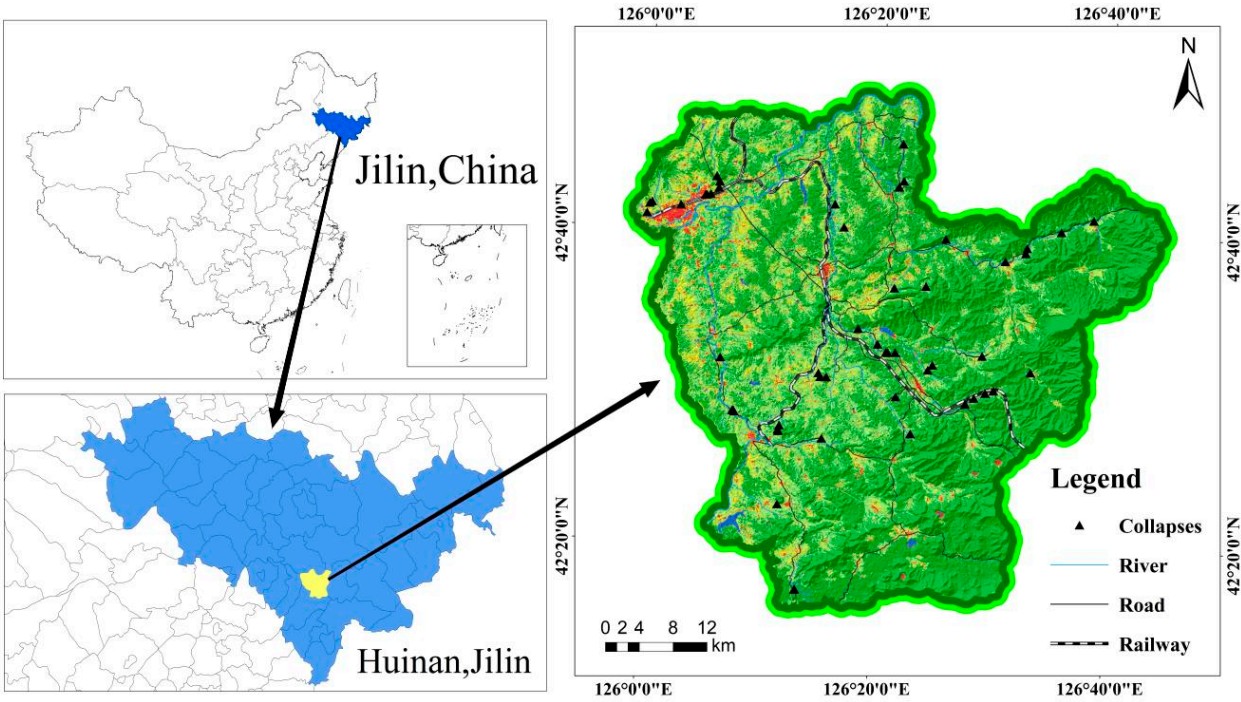

**Figure 1.** Location map of the study area.

*2.2. Collapse Evaluation Indicators*

2.2.1. Establishment of the Indicator System

This study is based on the evaluation indicators selected from the formation principle of natural disaster risk. For disaster hazards, in addition to reviewing the relevant literature, we also combined the geological characteristics of the study area and the mechanism of the occurrence of collapse hazards and filtered out the high-correlation evaluation indicators with the occurrence of collapses after comparison. For the exposure, vulnerability, emergency response, and recovery capability of the disaster acceptor, indicators related to population, economy, and education, as well as infrastructure, were considered from the perspective of human society and according to the population, economic, and social situation of the study area.

In summary, 21 evaluation indicators were selected for the study to apply to the collapse risk mapping, as follows:

Selection of Hazard Indicators

The hazard contains 11 indicators, namely slope angle, slope aspect, multi-year average precipitation, distance from the fault, distance from the river, distance from the road, landform type, vegetation type, NDVI, lithology, and mining point density. Slope angle is a crucial factor when examining the potential for collapses in an area, and where the slope angle is large, the risk of collapses is higher due to the increased gravitational effect

of the mountain and reduced resistance [29]. The slope aspect is one of the topographic factors, and differences in the slope aspect can result in differences in the direction of light exposure, light intensity, and hours of light on the slope [30]. Rainfall is a primary external factor of geological disasters in the region. Both surface water and groundwater's perennial scouring and erosion effects on geotechnical bodies will reduce the mechanical properties of geotechnical bodies [31], providing conditions for the formation of disasters such as collapse and landslides. The severity, distribution, and type of geological disasters are closely related to lithology, and lithology is an intrinsic factor in the development or occurrence of geological disasters [32]. The distance from the fault reflects the influence of geological formations on the formation and development of collapse disasters in many ways. The development of faults reduces the overall joint strength of the rock mass, resulting in a reduction in shear strength, which makes the rock mass vulnerable to destabilisation and damage under the action of external factors [18]. Distance from the river reflects the gradient relationship between rivers and the possible occurrence of collapses, with hills close to rivers being more prone to collapse disasters [25,33]. Distance from the road and mining point density [33] represent the intensity of human engineering activities, which can cause the movement of geotechnical bodies and groundwater transport under the influence of human activities and often cause geological disasters when beyond a stable state. Topography is the main controlling factor in determining the extent of geological disasters, and the presence of certain topographical conditions must be accompanied by a corresponding type of geological disaster. To a certain extent, a certain topography necessarily corresponds to a certain type of geological disaster. Lush vegetation can effectively reduce the degree of influence of the external environment on geological disasters [34,35], and geological disasters are less likely to be caused in low human activity areas with lots of vegetation. The spatial distribution of the 11 hazard indicators is shown in Figure 2.

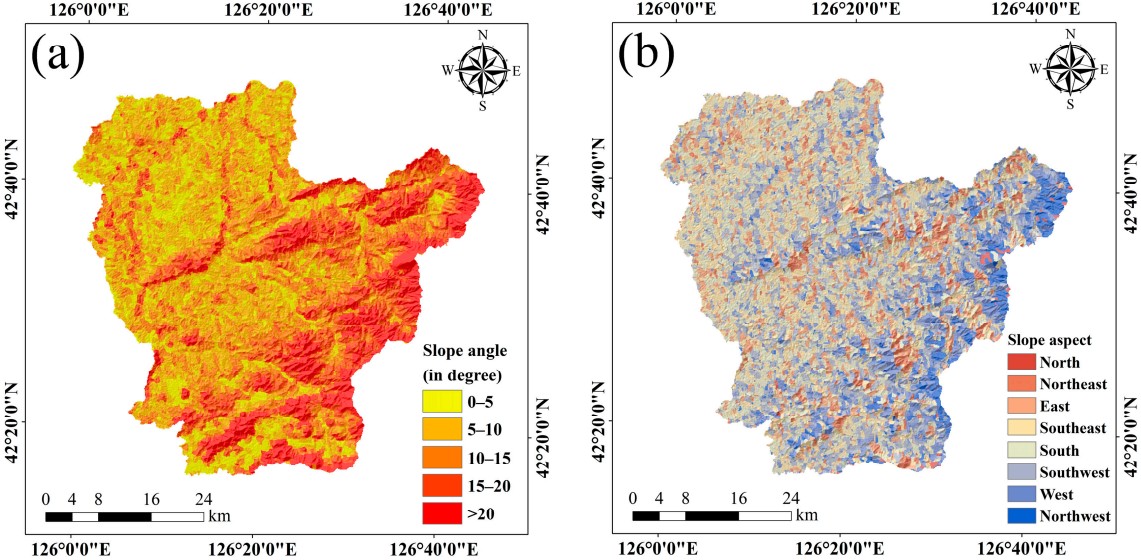

**Figure 2.** *Cont.*

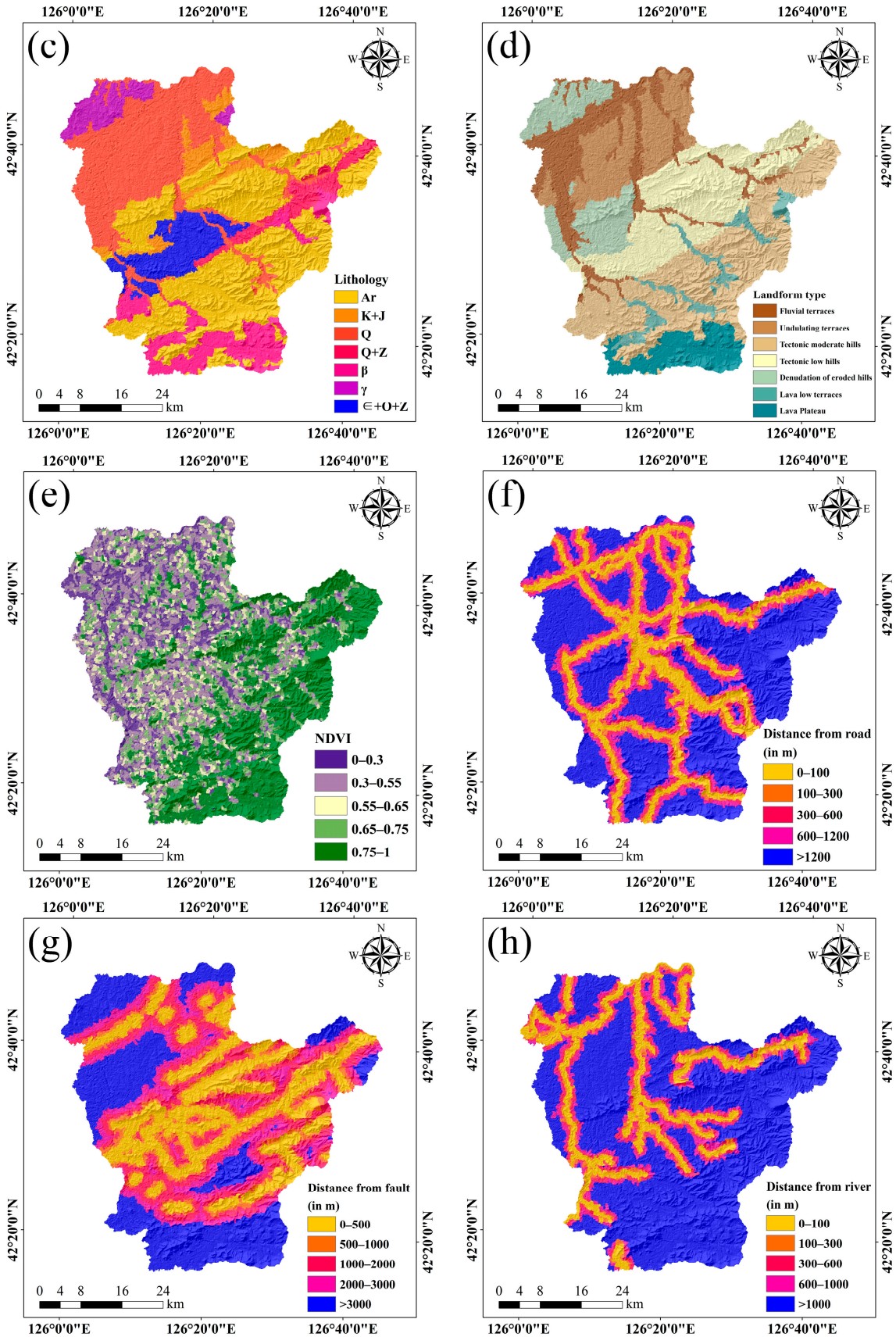

**Figure 2.** *Cont.*

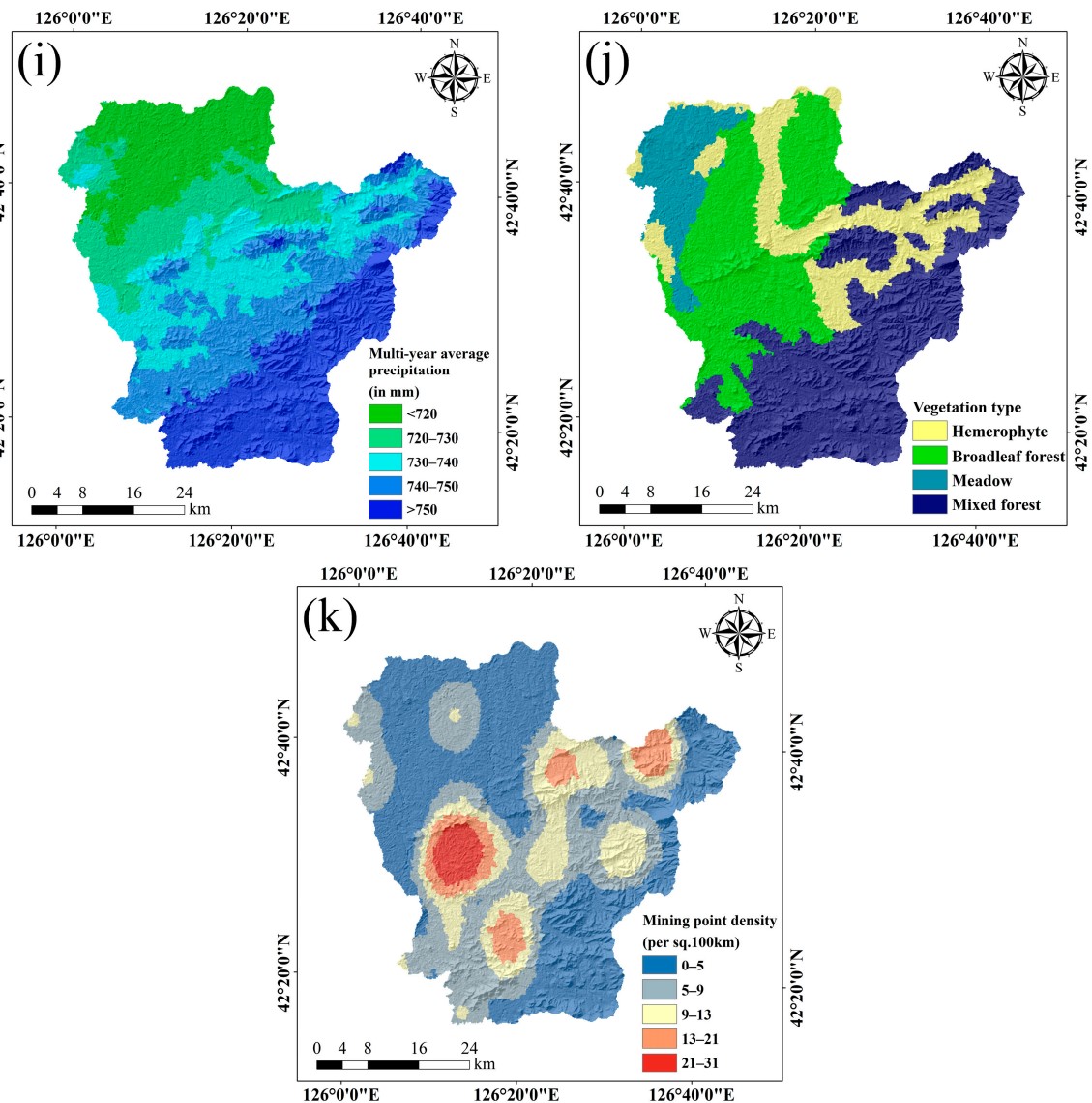

**Figure 2.** Hazard indicator maps: (**a**) slope angle; (**b**) slope aspect; (**c**) lithology; (**d**) landform type; (**e**) NDVI; (**f**) distance from road; (**g**) distance from fault; (**h**) distance from river; (**i**) multi-year average precipitation; (**j**) vegetation type; and (**k**) mining point density.

Selection of Exposure Indicators

Exposure refers to the economic, social, and natural environmental systems that are vulnerable to geological disasters, including agriculture, human living conditions, and the ecological environment. In this study, exposure mapping selected indicators such as population density, road density, housing density, and GDP are used. Population density is calculated by dividing the population of a township by its area. Areas with high population density have greater pressure on the living environment, leading to higher corresponding exposure [36], which is more severely affected by collapses. GDP is a measure of economic development [37], which is taken from a kilometre grid dataset of the spatial distribution of GDP in China, and the higher the GDP of the study area, the higher its exposure. Housing density and road density, which are key indicators of the urban development scale [38], are important factors regarding exposure, whereby dense housing and roads reflect greater exposure. The spatial distribution of the four exposure indicators is shown in Figure 3.

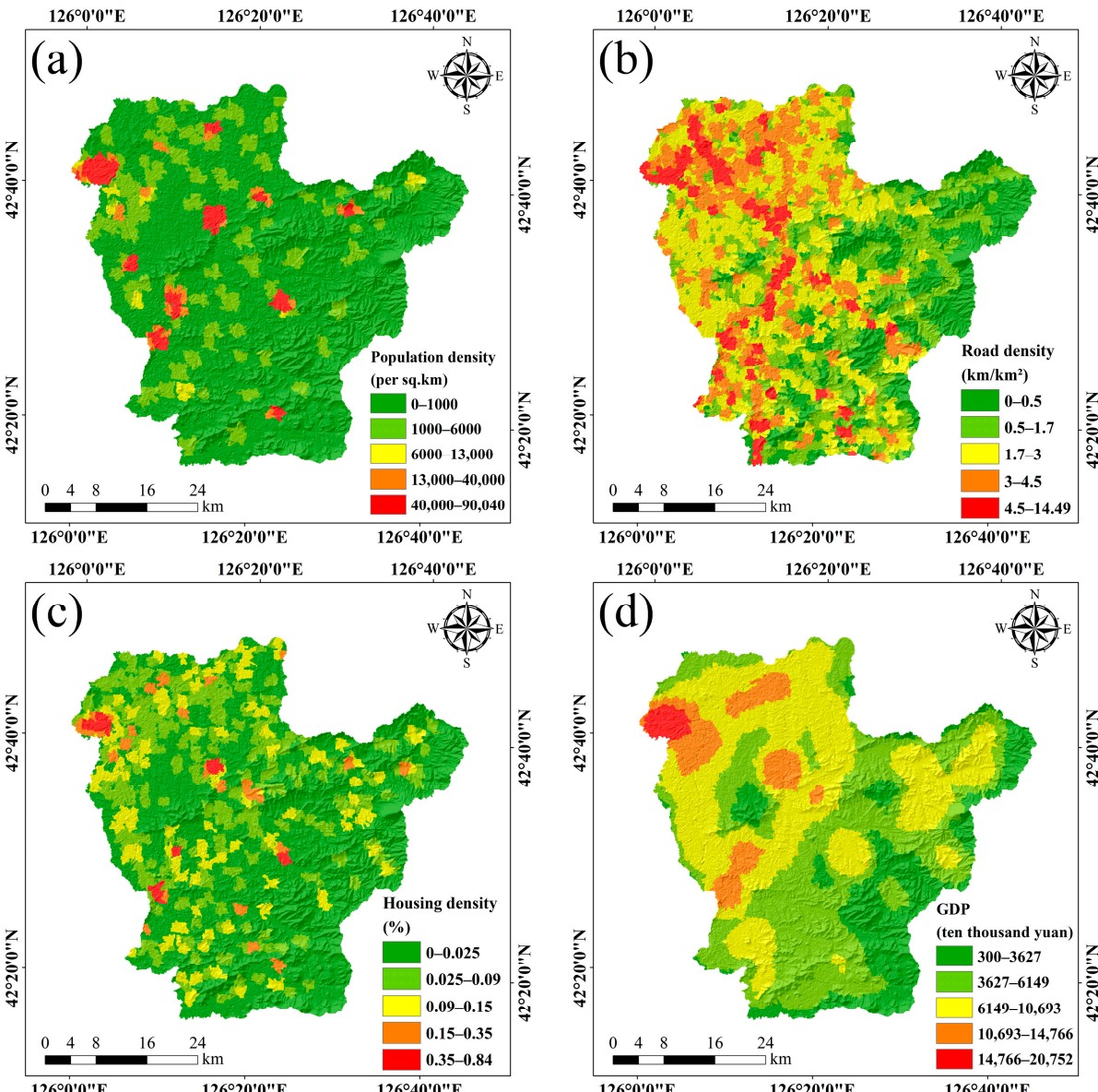

**Figure 3.** Exposure indicator maps: (**a**) population density; (**b**) road density; (**c**) housing density; and (**d**) GDP.

Selection of Vulnerability Indicators

Vulnerability refers to the degree of damage and loss to the disaster bearers in Huinan County due to potential disaster factors, including population, property, and ecosystems. The level of human vulnerability depends on factors such as age and awareness of disaster preparedness [28]. The selected vulnerability indicators in this study include the proportion of vulnerable populations, education status, and residential buildings. The vulnerable population includes people aged 0–17 and over 60 years old, as well as those with poor health who are more susceptible to disasters [28,38]. Education increases awareness of disaster preparedness, leading to reduced vulnerability. Residential buildings are human property and are more prone to collapse disasters when located in close proximity to mountainous areas [37]. The spatial distribution of these three vulnerability indicators is displayed in Figure 4.

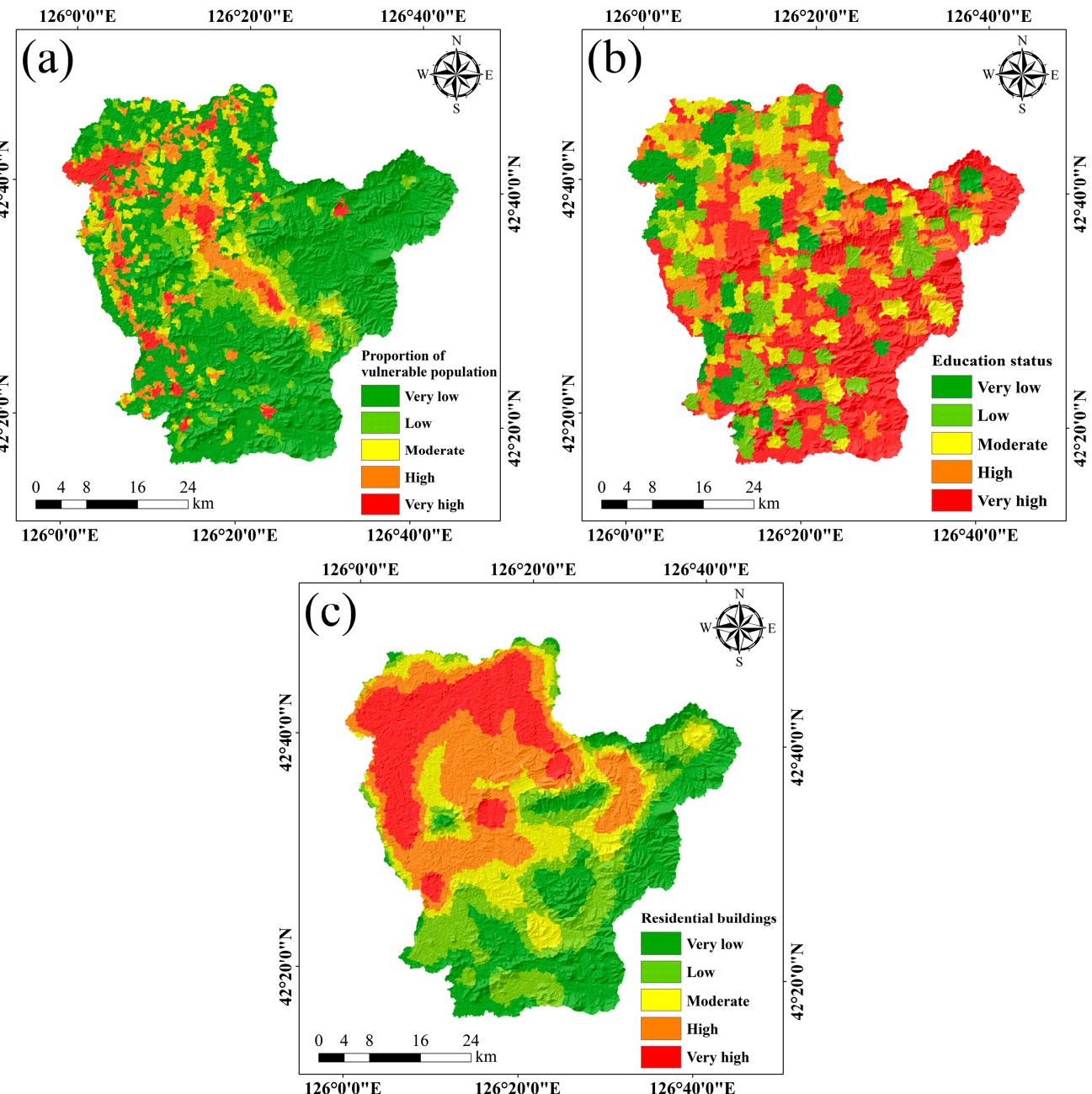

**Figure 4.** Vulnerability indicator maps: (**a**) proportion of vulnerable population; (**b**) education status; and (**c**) residential buildings.

Selection of Emergency Response and Recovery Capability Indicators

Emergency response and recovery capability refer to a series of disaster prevention measures taken to reduce the damage caused when a disaster bearer is affected by a collapse disaster. The level of local economic development and the number of emergency response agencies reflect the level of emergency response and recovery capability. In this study, education investment, local financial revenue, and the capacity of relief agencies are selected as indicators of emergency response and recovery capability. The spatial distribution of these three indicators is presented in Figure 5.

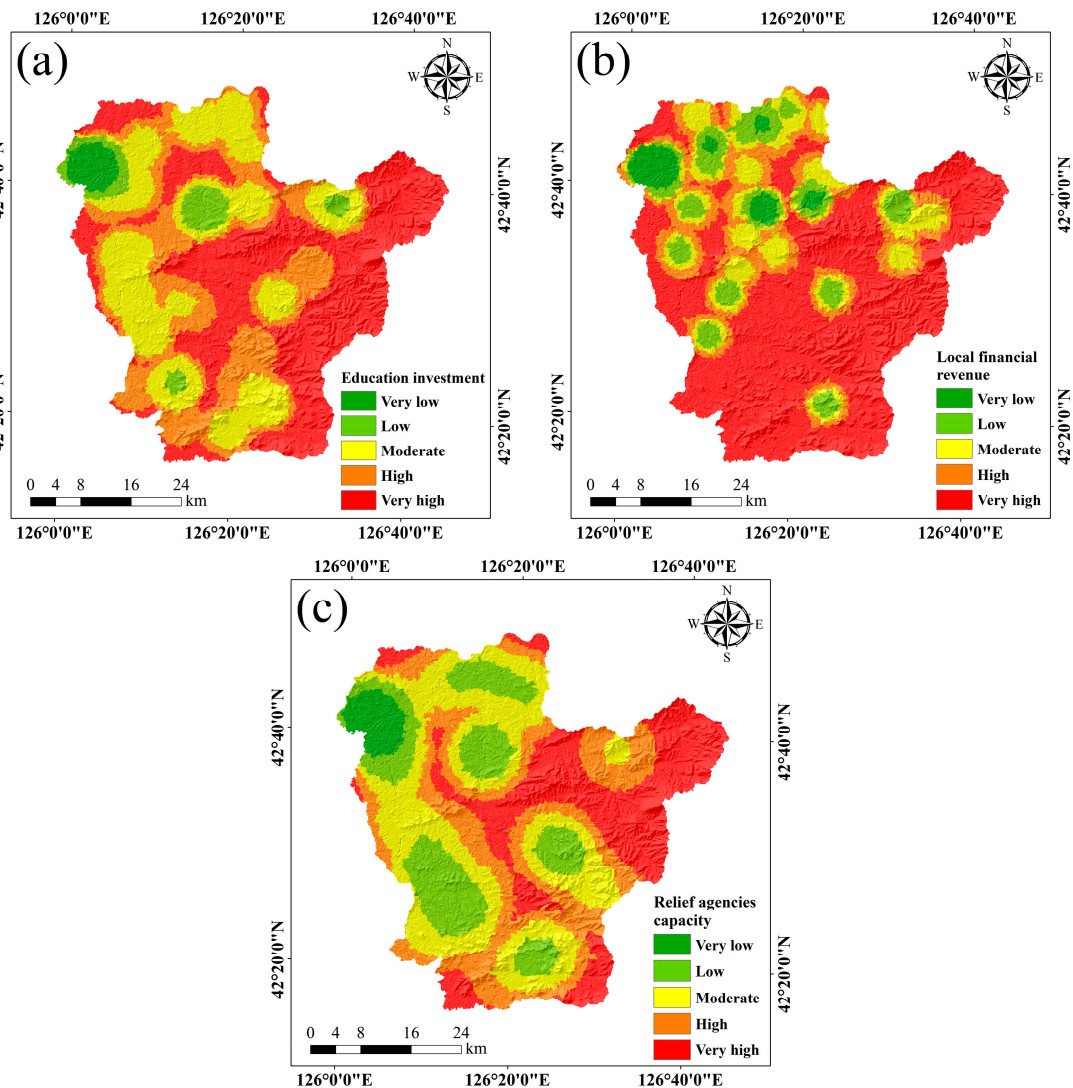

**Figure 5.** Emergency response and recovery capability indicator maps: (**a**) education investment; (**b**) local financial revenue; and (**c**) relief agencies' capacity.

### 2.2.2. Multicollinearity Analysis of Evaluation Indicators and Results

There may be statistical covariance between the initially selected collapse evaluation indicators, which can lead to the vulnerability model not being able to accurately analyse the true relationship between the evaluation indicators and collapses. Before building the model, this paper uses tolerance (TOL) and variance inflation factor (VIF) to test for factor covariance to ensure that the indicators are independent of each other [39]. The VIF is calculated as follows:

$$VIF = \frac{1}{1 - R_i^2} \tag{1}$$

where $R_i$ is the negative correlation coefficient of an evaluation indicator to other evaluation indicators. It is generally accepted that when the *VIF* value is greater than 10, there is strong co-collinearity between the factors, and the variable should be removed from the model.

Multiple covariance analysis was performed on the evaluation indicators by *VIF* and TOL to ensure that each evaluation indicator was independent of the other, and if the *VIF* of an evaluation indicator was greater than 10, it was excluded. As can be seen from the results in Table 1, the *VIF* values of all indicators were less than 10, indicating that each indicator was independent of the other and did not need to be excluded.

**Table 1.** Variance inflation factor (VIF) values and tolerance (TOL) for evaluation indicators.

| Indicator | TOL | VIF |
|---|---|---|
| Local financial revenue | 0.16 | 6.248 |
| Distance from road | 0.179 | 5.602 |
| Multi-year average precipitation | 0.266 | 3.755 |
| Vegetation type | 0.34 | 2.942 |
| Education investment | 0.351 | 2.849 |
| Slope aspect | 0.369 | 2.707 |
| Lithology | 0.369 | 2.712 |
| Education status | 0.432 | 2.312 |
| Population density | 0.477 | 2.095 |
| Landform type | 0.56 | 1.787 |
| Road density | 0.578 | 1.731 |
| Distance from river | 0.606 | 1.65 |
| Proportion of vulnerable population | 0.627 | 1.595 |
| Housing density | 0.75 | 1.333 |
| GDP | 0.796 | 1.256 |
| Relief agencies' capacity | 0.8 | 1.251 |
| NDVI | 0.846 | 1.182 |
| Slope angle | 0.898 | 1.114 |
| Distance from fault | 0.906 | 1.104 |
| Mining point density | 0.913 | 1.095 |
| Residential buildings | 0.947 | 1.055 |

*2.3. Data Collection*

This study constructs a collapse risk mapping indicator system for Huinan County from four elements: hazard, exposure, vulnerability, and emergency response and recovery capability. The data mainly include vector and raster data from remote sensing, meteorology, and basic geographic information, as well as attribute data related to population and economy. The data for lithology, distance from fault, landform type, mining point density, housing density, road density, and residential buildings were obtained from the Report on Geological Disaster Investigation and Mapping in Huinan County, Jilin Province, where the distance from fault was processed using the Euclidean Distance, the mining point density and residential buildings were processed using the Kernel Density, and the housing density and road density were processed through the Calculate Geometry of the Attribute Table. The slope angle and slope aspect are derived from DEM digital elevation data (ASTER GDEM 30 M) from the Geospatial Data Cloud, which can be extracted directly by ArcGIS 10.8.1. The data on distance from the river and distance from the road were obtained from the National Catalogue Service for Geographic Information and then processed using Euclidean Distance. Multi-year average precipitation was obtained directly from the Resource and Environment Science and Data Center of the Chinese Academy of Science. Vegetation type was obtained from the databox, and NDVI data were obtained from Landsat 8 OLI_TIRS in the Geospatial Data Cloud, then processed and converted to raster data using ENVI 5.3 software. Population density and proportion of vulnerable populations are available directly from the Worldpop Hub, with a resolution of 100 m selected. GDP data are available directly from the National Earth System Science Data Center, with a resolution of 1 km. Data on education status, education investment, local financial revenue, and relief agencies' capacity were obtained from POI from Planning Cloud (http://guihuayun.com/ (accessed on 15 February 2023)) for point extraction, and then Kernel Density was processed using ArcGIS 10.8.1, while the data used for analysing these evaluative indicators were obtained from the Tonghua Statistical Yearbook. Table 2 shows the selection of each indicator and its source, and Figure 6 shows the technical route and methodology.

**Table 2.** Indicators selected for the collapse risk model and their sources.

| Sl. No. | Indicator | Data Types | Resolution/Year | Data Source |
|---|---|---|---|---|
| | Hazard indicators | | | |
| 1 | Lithology | Raster data | 1:50,000 | Report on Geological Disaster Investigation and Mapping in Huinan County, Jilin Province |
| 2 | Distance from fault | Vector data | 1:50,000 | Report on Geological Disaster Investigation and Mapping in Huinan County, Jilin Province |
| 3 | Slope angle | Raster data | 30 m | https://www.gscloud.cn (accessed on 30 February 2023) |
| 4 | Slope aspect | Raster data | 30 m | https://www.gscloud.cn (accessed on 30 February 2023) |
| 5 | Landform type | Vector data | 1:50,000 | Report on Geological Disaster Investigation and Mapping in Huinan County, Jilin Province |
| 6 | Distance from river | Vector data | 1:1,000,000 | https://www.webmap.cn/ (accessed on 9 March 2023) |
| 7 | Multi-year average annual precipitation | Raster data | 1 km | https://www.resdc.cn/ (accessed on 15 March 2023) |
| 8 | Vegetation type | Vector data | 1:1,000,000 | https://www.databox.store (accessed on 11 February 2023) |
| 9 | NDVI | Raster data | 30 m | Landsat 8 OIL_TIRS |
| 10 | Distance from road | Vector data | 1:1,000,000 | https://www.webmap.cn/ (accessed on 9 March 2023) |
| 11 | Mining point density | Vector data | 1:50,000 | Report on Geological Disaster Investigation and Mapping in Huinan County, Jilin Province |
| | Exposure indicators | | | |
| 1 | Population density | Raster data | 100 m | https://www.worldpop.org/ (accessed on 14 February 2023) |
| 2 | Housing density | Vector data | 1:50,000 | Report on Geological Disaster Investigation and Mapping in Huinan County, Jilin Province |
| 3 | Road density | Vector data | 1:50,000 | Report on Geological Disaster Investigation and Mapping in Huinan County, Jilin Province |
| 4 | GDP | Raster data | 1 km | http://www.geodata.cn/ (accessed on 16 March 2023) |
| | Vulnerability indicators | | | |
| 1 | Proportion of vulnerable population | Raster data | 100 m | https://www.worldpop.org/ (accessed on 14 February 2023) |
| 2 | Education status | Attribute data | 2015–2019 | Tonghua Statistical Yearbook |
| 3 | Residential buildings | Vector data | 1:50,000 | Report on Geological Disaster Investigation and Mapping in Huinan County, Jilin Province |
| | Emergency response and recovery capability indicators | | | |
| 1 | Education investment | Attribute data | 2015–2019 | Tonghua Statistical Yearbook |
| 2 | Local financial revenue | Attribute data | 2015–2019 | Tonghua Statistical Yearbook |
| 3 | Relief agencies' capacity | Attribute data | 2015–2019 | Tonghua Statistical Yearbook |

All data for this study were processed in ArcGIS 10.8.1, and the geographical coordinates were GCS_WGS_1984. The data analysis was carried out in the projection coordinates of WGS 1984 UTM Zone 52 N, and the cell size of all indicators was adjusted to 100 m when doing the analysis, so the total number of cells for each indicator map was 227,121.

### 2.4. Mapping Unit

The division of mapping units is the first and most important basis for the evaluation of the susceptibility, hazard, and risk of disasters in large-scale areas [40]. Its purpose is to extract the actual area into the unit, which in turn makes the results of the collapse risk mapping more accurate. Before doing the analysis, the data of all evaluation indicators are extracted into the mapping unit, and then GIS technology is used to conduct spatial overlay analysis of all evaluation indicators and visualise the data so that the results obtained will

be more representative of the corresponding regions in reality. There are five main types of mapping units: grid unit, regional unit, watershed unit, uniform condition unit, and slope unit [41,42].

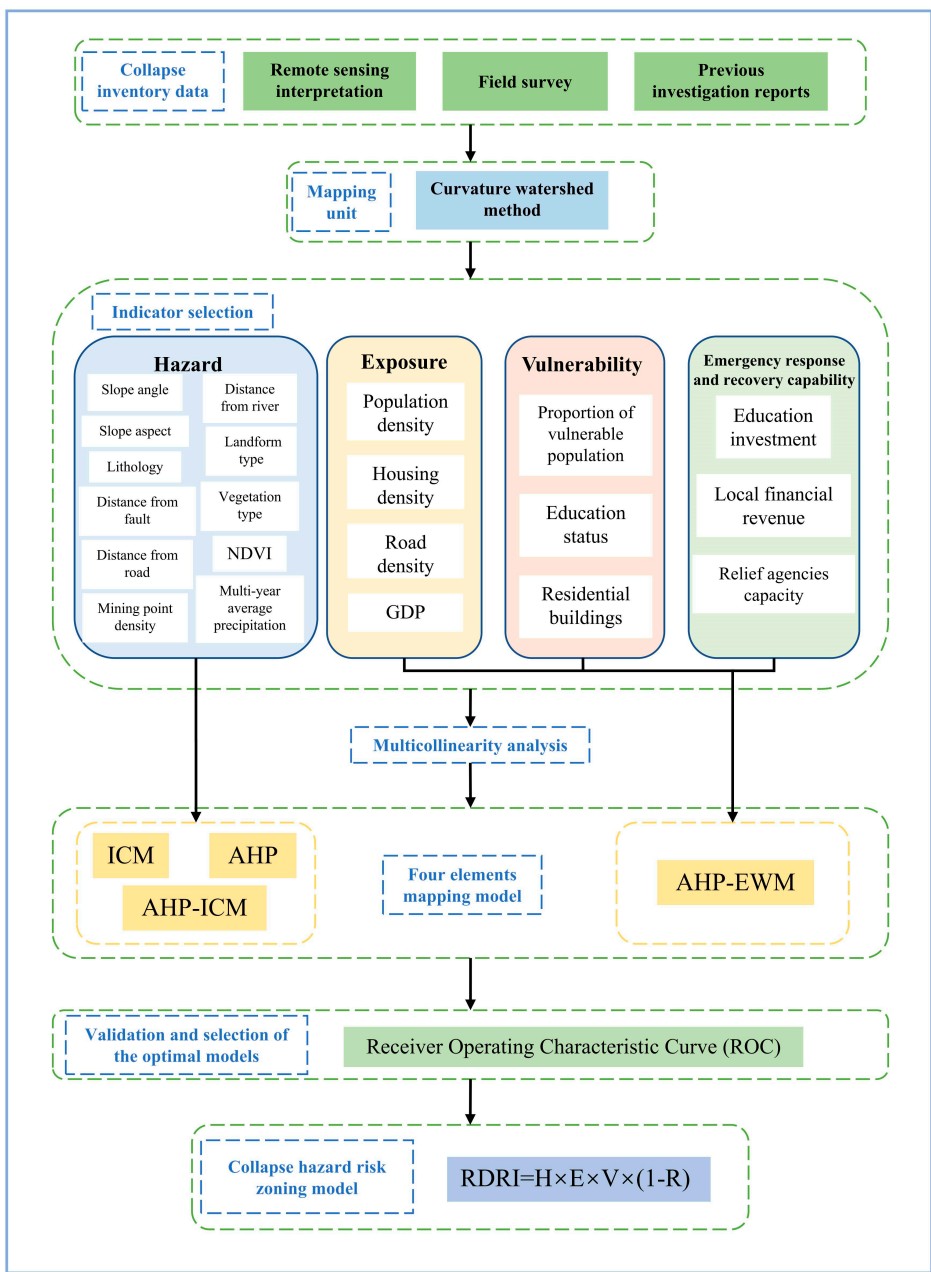

**Figure 6.** Flow chart of collapse risk mapping.

The accuracy of the mapping unit delineation reflects the fact that the study area is delineated as accurately as possible on ArcGIS so that a more accurate analysis of the study area can be carried out. Therefore, the correct choice of mapping unit delineation is essential for collapse risk evaluation. Previously, grid units were chosen as mapping units for collapse disaster evaluation [43]; however, the disadvantage of this approach is that it does not reflect the geological characteristics of the study area, which is divided into a certain number of regular grids, and after extracting information about the study area onto grid units, the unit is almost completely irrelevant to information about the spatial topography of the study area. Therefore, the slope unit was selected as the mapping unit in this paper, and it was divided into slope units using the hydrological analysis module in

ArcGIS 10.8.1 for the 90 m resolution DEM data of the study area. Currently, slope units include the hydrological analysis method and the curvature watershed method [18,44]. The hydrological analysis method uses DEM data with inverted DEM data to extract ridge lines and valley lines in the study area. The disadvantages of this method are the complexity of the operation, the need to control the appropriate flow thresholds and the resolution of the DEM, and the difficulty of distinguishing between horizontal and sloping terrain. The curvature watershed method is through the plane curvature and profile curvature of great and small values, reflecting the changes in slope and slope direction, and thus can identify the ridge line and valley line well, including the horizontal terrain and inclined terrain between the boundary; the method is not only simple to operate, but in the later stages of manual modification of the unreasonable units, greatly reduces the amount of work. The delineation accuracy is only related to the resolution of the DEM [25]. In this paper, the curvature watershed method is chosen to divide the study area into slope units. Figure 7 shows the detailed slope unit division process, and Figure 8 shows the slope unit delineation results of the study area. After comparing the results based on the slope unit division with the real terrain, the study area was divided into 10,096 units, and the geometric calculations yielded an area of 1.37 km$^2$ for the largest unit divided and 0.005 km$^2$ for the smallest unit.

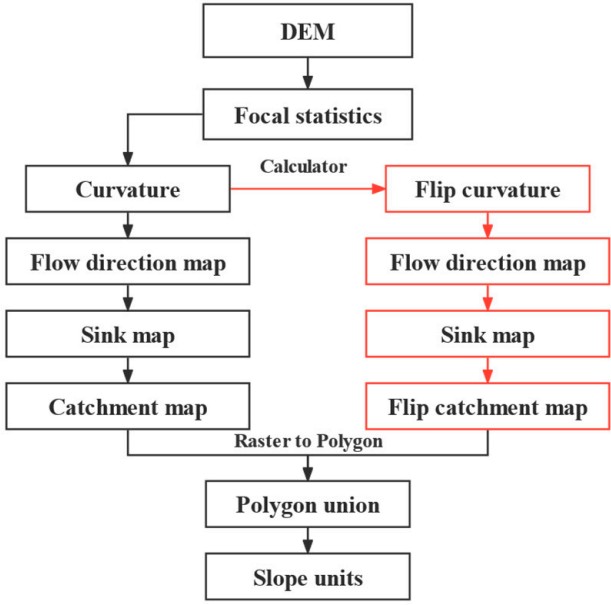

**Figure 7.** Slope unit classification process based on the curvature watershed method.

*2.5. Collapse Inventory*

The precise location and extent of the collapse are crucial in the process of collapse risk mapping [45]. For any collapse risk mapping, one of the important processes is collapse inventory, which relates to the location, number, extent of impact, and intensity of activity of collapses in an area. In this paper, remote sensing interpretation, field investigation, and synthesis of previous survey information are used to create a collapse inventory map. Information on collapses in the study area was obtained from the geological disaster survey and mapping report of Huinan County, Jilin Province. A total of 52 collapses were identified in this study using available information (Figure 8). According to statistics, nearly half of the collapse disaster potential sites (20) in the study area have formed multiple mountain-cut slopes due to the construction of roads and the intense folk mining activities in and around the chert and limestone quarries.

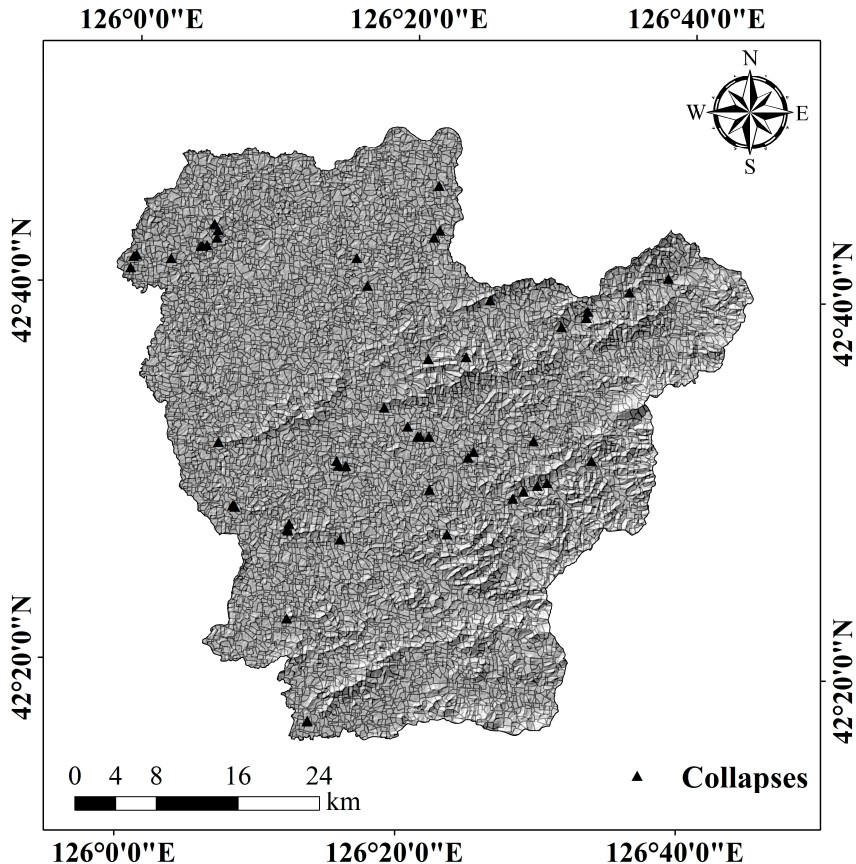

**Figure 8.** Slope unit division and collapse inventory in the study area.

*2.6. Collapse Mapping Model*

2.6.1. Hazard Mapping Model

Information Content Model (ICM)

The *ICM* is one of the basic concepts of information theory, which was proposed by Claude E. Shannon. According to the information theory, the amount of information can be measured by probability. *ICM* is a statistical analysis and prediction method that has been widely used in geohazard risk assessment [17,18,46]. The method calculates the *ICM* value of each indicator based on known information about the collapse and its evaluation indicators; the smaller the *ICM* value, the less likely it is to cause collapse disasters. The formula is calculated as follows:

$$I(X_i, H) = \ln \frac{N_i/N}{S_i/S} \tag{2}$$

where $X_i$ is the evaluation indicator, $I(X_i, H)$ is the information value of each evaluation indicator $X_i$ affecting the occurrence of collapse, $N$ is the total number of collapse disasters in the study area, $N_i$ is the number of collapses contained in each evaluation indicator $X_i$, $S$ is the total number of rasters in the study area, and $S_i$ is the total number of rasters in each evaluation indicator $X_i$. Then, the *ICM* values were assigned to each evaluation indicator map to reclassify them. Finally, the collapse hazard index (*CHI*) was obtained by superimposing each evaluation indicator map. The calculation formula is as follows:

$$CHI_{ICM} = \sum ICM_i \tag{3}$$

where *ICM* refers to the evaluation indicator map reclassified by the informative values, and *i* denotes hazard indicators from the 1st to the 11th.

Analytical Hierarchy Process (AHP)

AHP is a multi-criteria decision-making method proposed by Thomas L. Saaty [47]. The weights of the indicators are determined by building a hierarchical model, decomposing them into a series of levels and criteria, and then determining the relative importance between the elements of each level by comparing them two by two. The relative importance of these indicators is assessed on a scale of 1–9 from low to high. The AHP method consists of three steps: firstly, constructing an affiliation model for the highest level of decision making (target level), the middle level (criterion level), and the bottom level (indicator level); secondly, establishing the corresponding judgement matrix; and thirdly, calculating the weights of each factor and testing the random consistency of each factor [20,48,49].

Establishing a system hierarchy: According to the specific performance and causes of collapse disasters in Huinan County, reasonable evaluation indicators were selected after combining the relevant situations in the study area with the analysis of the research results of Huinan County, analysing the main influencing elements of collapse disasters occurring in the study area, such as the natural environment and geological elements, etc. The hierarchy model and specific factors were built in the process of the study (Figure 9).

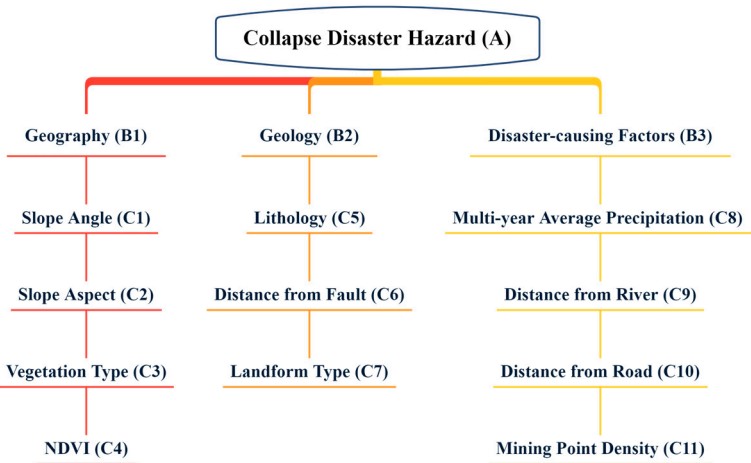

**Figure 9.** Schematic diagram of the structure of the AHP of collapse disaster.

Establishing a judgement matrix: The structural model of the collapse disaster analytical hierarchy process constructed in Figure 9 allows for the analysis of factors and the accurate representation of the affiliation problem of factors. In accordance with the 1~9 scale method and other relevant contents, there will be differences in the hierarchical aspects between the factors. After specifying their weights, the judgement matrices are built for the A–B and B–C layers, respectively. The judgement matrix can be expressed as follows:

$$X = (x_{ij})_{n \times n} = \begin{bmatrix} x_{11} & x_{12} & \dots & x_{1n} \\ x_{21} & x_{22} & \dots & x_{2n} \\ \dots & \dots & \dots & \dots \\ x_{n1} & x_{n2} & \dots & x_{nn} \end{bmatrix} \tag{4}$$

where $X$ is the judgement matrix; $x_{ij}$ is the result of the comparison of the importance of factor $i$ and factor $j$, and has the following property:

$$a_{ij} = \frac{1}{a_{ji}} \tag{5}$$

Relative weights and consistency test: In order to check the reasonableness of the weight determination method, a consistency test is required. The maximum eigenvalue

$\lambda$max of the matrix is utilised while the corresponding eigenvectors are obtained, and the consistency of the judgment matrix is verified according to Equations (6) and (7).

$$CI = \frac{\lambda_{\max} - n}{n - 1} \tag{6}$$

$$CR = \frac{CI}{RI} \tag{7}$$

where *CI* is the consistency indicator, and the smaller the value, the lower the degree of inconsistency; $\lambda$max is the maximum eigenvalue of the judgment matrix; *n* is the order; *CR* is the consistency ratio; and *RI* is the random consistency indicator.

Based on the principles of *AHP*, the relative weights of each hazard indicator are calculated, and the collapse hazard index (*CHI*) is calculated as follows:

$$CHI_{AHP} = \sum_{i=1}^{n} (W_i \times W_{ij}) \tag{8}$$

where $W_i$ is the weight of evaluation index *i*, $W_{ij}$ is the *i*-th evaluation indicator and the *j*-th sub-class weight, and *n* is the number of hazard evaluation indicators, with *n* = 11 in this study.

AHP-ICM Model

*AHP-ICM* is a combination of subjective and objective methods, which avoids the limitations of a single method and improves the comprehensiveness and accuracy of collapse hazard evaluation. This method has been widely used in the risk evaluation of geologic disasters [18,19,24,50]. After determining the sub-classes to which the hazard evaluation indicators belong, the information value of each sub-class is calculated using the *ICM*, and the weights of each evaluation indicator are obtained using the *AHP*, and the two are coupled for calculation. In this study, the *AHP-ICM* model is used for collapse hazard calculation, and the specific formula is

$$CI_{ij} = W_i \times I_{ij} \tag{9}$$

where $CI_{ij}$ is the combined information value of the sub-class *j* in evaluation indicator *i*, $W_i$ is the weight of each evaluation indicator obtained by *AHP*, and $I_{ij}$ is the information value of the *i*th evaluation indicator and the *j*th sub-class.

2.6.2. Exposure, Vulnerability, and Emergency Responses and Recovery Capability Mapping Model
Entropy Weighting Method (EWM)

The entropy weighting method is an objective weight-defining method that is used to characterise the relative intensity of each indicator in the evaluation system of a target system. The entropy method does not have any subjective analysis for the calculation of individual indicator weights, relying only on objective indicator data to calculate the magnitude of the indicator weights [24,51]. The specific steps are as follows:

With *m* objects, each with *n* evaluation indicators, *a* judgment matrix is constructed.

$$A = (a_{ij})_{m*n} (i = 1, 2, \ldots, m; j = 1, 2, \ldots, n) \tag{10}$$

where $a_{ij}$ is the value of the *j*th indicator for the *i*-th object.

The judgment matrix is normalised to obtain the normalised judgment matrix $D = (d_{ij})_{mn}$.

$$d_{ij} = \frac{r_{ij} - r_{\min}}{r_{\max} - r_{\min}} \tag{11}$$

$$d_{ij} = \frac{r_{\max} - r_{ij}}{r_{\max} - r_{\min}} \qquad (12)$$

where $r_{\max}$ is the highest indicator value under different objects for the same indicator and $r_{\min}$ is the lowest indicator value under different objects for the same indicator.

For a situation with m objects and *n* indicators, the entropy of the evaluation indicator can be determined as

$$S_j = -\frac{\sum_{i=1}^{m} f_{ij} \ln f_{ij}}{\ln m} \ (i = 1, 2, \ldots, m; j = 1, 2, \ldots, n) \qquad (13)$$

where $f_{ij} = \frac{1 + d_{ij}}{\sum_{i=1}^{m} (1 + d_{ij})}$.

The entropy weight vector *W* of the evaluation indicators is

$$W = (\omega_j)_{1 \times n} \qquad (14)$$

where $\omega_j$ is the entropy weight of the *j*th evaluation indicator, and $\omega_j = \frac{1 - S_j}{n - \sum_{j=1}^{n} S_j}$.

AHP-EWM Model

Firstly, the sub-classes of exposure, vulnerability, and emergency response and recovery capability evaluation factors are determined; their weights are calculated using the AHP, then the weights of the evaluation factors are calculated using the EWM, and finally, the two are coupled to derive the combined weights of each sub-class in each evaluation factor. In this study, the method is used for the calculation of exposure, vulnerability, and emergency response and recovery capability. The specific formulae are as follows:

$$S_{ij} = W_{ij} \times W_e \qquad (15)$$

where $S_{ij}$ is the combined weight value of sub-class *j* in evaluation indicator *i*. o is the *i*-th evaluation indicator and the weight of the *j*-th sub-class calculated by hierarchical analysis. *r* is the weight of each evaluation indicator calculated by the entropy weight method.

2.6.3. Collapse Risk Mapping Model

In this study, using the formation principle of natural disaster risk, the four elements of geohazard disaster, the selected index system, and the integrated weighted analysis method were used to establish the collapse disaster risk index model, whose formula can be expressed as follows [28]:

$$RDRI = H \times E \times V \times (1 - R) \qquad (16)$$

where *RDRI* represents the collapse disaster risk index; a higher value means a higher level of collapse disaster risk. *H*, *E*, *V*, and *R* represent the collapse disaster hazard, exposure, vulnerability, and emergency response and recovery capability indicator indices, respectively.

## 3. Results and Analysis of the Hazard Mapping Model

### 3.1. Results of the Model

3.1.1. Results of the Information Content Model (ICM)

There is a close relationship between the collapse disaster and the information content (IC) value; when IC > 0, the probability of collapse is greater than the average value, i.e., the probability of collapse is high; conversely, the probability of collapse is low in the reverse case; if IC = 0, the probability of occurrence is of an average value. The results are shown in Table 3.

**Table 3.** Spatial relationship between each collapse evaluation indicator and collapse.

| Indicator | Class | Collapse Count | Total Count | ICM |
|---|---|---|---|---|
| Slope angle | 0–5 | 5 | 32,898 | −0.4099 |
| | 5–10 | 12 | 68,537 | −0.2682 |
| | 10–15 | 20 | 51,643 | 0.5256 |
| | 15–20 | 12 | 38,386 | 0.3115 |
| | >20 | 3 | 35,653 | −1.0010 |
| Slope aspect | North | 0 | 1296 | 0.0000 |
| | Northeast | 1 | 12,326 | −1.0375 |
| | East | 2 | 25,050 | −1.0535 |
| | Southeast | 14 | 48,,583 | 0.2300 |
| | South | 12 | 51519 | 0.0172 |
| | Southwest | 19 | 46,038 | 0.5892 |
| | West | 3 | 30,587 | −0.8477 |
| | Northwest | 1 | 11,718 | −0.9869 |
| Multi-year average precipitation | <720 | 5 | 39,165 | −0.5841 |
| | 720–730 | 11 | 35,490 | 0.3029 |
| | 730–740 | 21 | 44,830 | 0.7159 |
| | 740–750 | 10 | 48,315 | −0.1009 |
| | >750 | 5 | 59,321 | −0.9993 |
| Lithology | Q | 6 | 62,666 | −0.8718 |
| | $\gamma$ | 11 | 10,226 | 1.5472 |
| | K + J | 3 | 12,000 | 0.0879 |
| | Ar | 10 | 90,418 | −0.7276 |
| | Q + Z | 2 | 10,027 | −0.1379 |
| | $\beta$ | 5 | 23,725 | −0.0829 |
| | $\in$ + O + Z | 15 | 18,059 | 1.2886 |
| Distance from fault | 0–500 | 18 | 56,250 | 0.3348 |
| | 500–1000 | 10 | 27,686 | 0.4559 |
| | 1000–2000 | 7 | 44,551 | −0.3765 |
| | 2000–3000 | 11 | 28,746 | 0.5136 |
| | >3000 | 6 | 69,888 | −0.9809 |
| Landform type | Fluvial terrace | 6 | 34,499 | −0.2749 |
| | Undulating terrace | 5 | 30,078 | −0.3201 |
| | Denudation of eroded hill | 9 | 23,872 | 0.4988 |
| | Tectonic low hill | 22 | 49,696 | 0.6594 |
| | Tectonic moderate hill | 6 | 61,991 | −0.8610 |
| | Lava low terrace | 2 | 10,751 | −0.2076 |
| | Lava plateau | 2 | 16,234 | −0.6197 |
| Distance from river | 0–100 | 32 | 33,744 | 1.4212 |
| | 100–300 | 3 | 10,748 | 0.1981 |
| | 300–600 | 7 | 17,049 | 0.5841 |
| | 600–1000 | 6 | 19,992 | 0.2707 |
| | >1000 | 4 | 145,588 | −2.1202 |
| Distance from road | 0–100 | 33 | 44,349 | 1.1787 |
| | 100–300 | 3 | 15,005 | −0.1355 |
| | 300–600 | 4 | 19,397 | −0.1046 |
| | 600–1200 | 7 | 33,720 | −0.0979 |
| | >1200 | 5 | 114,650 | −1.6582 |
| Vegetation type | Hemerophyte | 18 | 42,454 | 0.6162 |
| | Broadleaf forest | 15 | 69,491 | −0.0589 |
| | Meadow | 7 | 21,029 | 0.3742 |
| | Mixed forest | 12 | 94,147 | −0.5857 |
| NDVI | 0–0.3 | 20 | 32,809 | 0.9793 |
| | 0.3–0.55 | 4 | 24,824 | −0.3513 |
| | 0.55–0.65 | 21 | 50,226 | 0.6022 |
| | 0.65–0.75 | 4 | 38,058 | −0.7786 |
| | 0.75–1 | 3 | 81,204 | −1.8241 |

**Table 3.** *Cont.*

| Indicator | Class | Collapse Count | Total Count | ICM |
|---|---|---|---|---|
| | 0–5 | 14 | 104,598 | −0.5369 |
| | 5–9 | 15 | 66,194 | −0.0103 |
| Mining point density | 9–13 | 14 | 35,987 | 0.5301 |
| | 13–21 | 7 | 14,501 | 0.7459 |
| | 21–31 | 2 | 5830 | 0.4044 |
| Total area | | 52 | 227,121 | |

As can be seen from Table 3, the ICM values for slope angles of 10–15° and 15–20° are 0.5256 and 0.3115, respectively, and according to previous studies, the higher the slope angle, the greater the probability of collapse occurring [29]. According to field surveys, within these two gradations, collapse is more likely to occur because of the steep cliffs or steep slopes within a small area formed by road cutting and engineering construction. The ICM values are 0.23 and 0.5892 for southeast- and southwest-facing slopes, respectively. For the multi-year average precipitation, the ICM values are 0.3029 and 0.7159 for 720–730 mm and 730–740 mm, respectively, with the higher precipitation concentrated in areas of the study area with less human activity, where the geotechnical structure is intact, the vegetation cover is larger and subject to precipitation erosion, and the erosion effect is less. For lithology, the ICM values for the granite rock group ($\gamma$) and the hard limestone rock group ($\in + O + Z$) are 1.5472 and 1.2886, respectively. As both $\gamma$ weathering fissures and primary fissures are more developed, the upper part is strongly weathered and the rock integrity is poor, while $\in + O + Z$ consists of medium-thick laminated pure limestone, sandstone and siltstone, with weakly developed karst, generally dominated by solution gaps and small caves, and in micro-landscapes of mostly steep cliffs and steep slopes, so these two rock groups are most prone to collapse. The degree of development of collapse disasters in Huinan County is obviously influenced by faults. Fractures are developed in the area, with more structural surfaces and locally weak surfaces or weak zones, with low rock strength and easy weathering and stripping. For landform type, the ICM values for denudation of eroded hill and tectonic low hill landform areas are 0.4988 and 0.6594, respectively. These two areas are more prone to collapse due to more residential settlements, developed transportation and strong human activities such as road cutting, deforestation, and mining. In mountainous areas close to rivers, the occurrence of collapses is mainly located within 100 m of the river due to the erosive action of the river. In this study area, as the construction of the road required extensive slope cutting, the collapse occurred mainly within 100 m of the road. The ICM values for hemerophyte and meadow were 0.6162 and 0.3742, respectively, with higher ICM values for NDVI within 0.65, indicating that areas with low vegetation cover are prone to collapse. Where the mining point density is high, collapse is more likely to occur due to the long-term disturbance of the geotechnical body. The results are shown in Figure 10a.

### 3.1.2. Results of the Analytical Hierarchy Process (AHP)

The degree of merit of the model was judged by CR based on the comparison between variables and indicators, and the consistency of the judgment matrix was accepted if CR < 0.1 [20,52–54]. Then, the weights Wi normalised by the criterion layer to the indicator layer (B–C) were obtained, and the weights of the target layer to the indicator layer (A–C) were obtained after normalising them again under the target layer (A), and since all CRs were less than 0.1, all judgment matrices met the consistency requirements and were considered to be reasonably weighted. The specific data content is shown in Tables 4 and 5. Based on the results, the multi-year average precipitation has the highest weighting of 0.4119, followed by distance from road and lithology at 0.1924 and 0.1118, respectively. This indicates that the multi-year average precipitation and distance from road and lithology are the main factors contributing to collapse in the study area. Based on Equation (8), the

AHP-generated collapse hazard map was derived by combining Tables 4 and 5 to calculate CHI, as shown in Figure 10b.

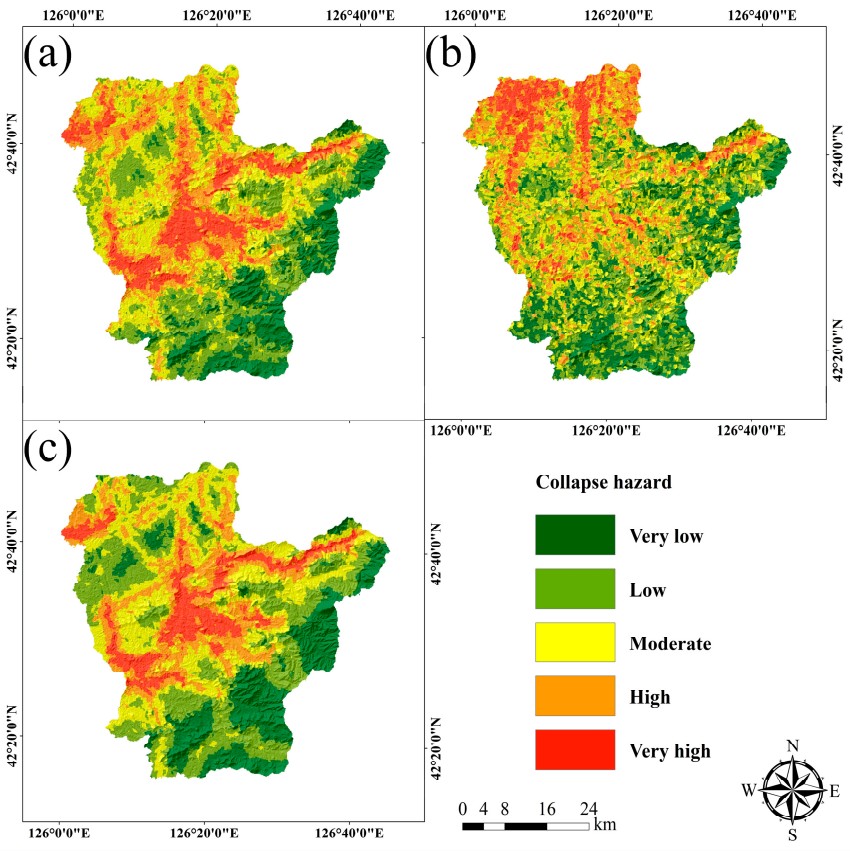

**Figure 10.** Results of hazard mapping: (**a**) information content method (ICM) method; (**b**) analytical hierarchy process (AHP) method; and (**c**) analytical hierarchy process–information content method (AHP-ICM) method.

**Table 4.** Judgement matrix of evaluation indicators and their weights.

| Target Layer (A) | Criterion Layer (B) | A–B Judgement Matrix | | | A–B Weight | Indicator Layer (C) | B–C Judgement Matrix | | | | B–C Weight | A–C Weight (Wi) |
|---|---|---|---|---|---|---|---|---|---|---|---|---|
| | Geography | 1 | 1/3 | 1/7 | 0.081 | Slope angle | 1 | 3 | 3 | 1/2 | 0.2947 | 0.0239 |
| | | | | | | Slope aspect | 1/3 | 1 | 1/3 | 1/5 | 0.0781 | 0.0063 |
| | | | | | | Vegetation type | 1/3 | 3 | 1 | 1/3 | 0.1537 | 0.0124 |
| | | | | | | NDVI | 2 | 5 | 3 | 1 | 0.1537 | 0.0383 |
| Collapse hazard (1.0000) | Geology | 3 | 1 | 1/5 | 0.1884 | Lithology | 1 | 3 | 3 | | 0.5936 | 0.1118 |
| | | | | | | Distance from fault | 1/3 | 1 | 1/2 | | 0.1571 | 0.0296 |
| | | | | | | Landform type | 1/3 | 2 | 1 | | 0.2493 | 0.047 |
| | Disaster-causing factors | 7 | 5 | 1 | 0.7306 | Multi-year average precipitation | 1 | 5 | 3 | 7 | 0.5638 | 0.4119 |
| | | | | | | Distance from river | 1/5 | 1 | 1/3 | 3 | 0.1178 | 0.0861 |
| | | | | | | Distance from road | 1/3 | 3 | 1 | 5 | 0.2634 | 0.1924 |
| | | | | | | Mining point density | 1/7 | 1/3 | 0.2 | 1 | 0.0550 | 0.0402 |

**Table 5.** Judgement matrix for evaluation indicators' sub-classes and their weights.

| Indicator | Class | Judgement Matrix | | | | | Weight (W_{ij}) |
|---|---|---|---|---|---|---|---|
| Slope angle | 0–5 | 1 | 1/3 | 1/3 | 1/4 | 1/3 | 0.06386 |
| | 5–10 | 3 | 1 | 1/3 | 1/3 | 1/3 | 0.10497 |
| | 10–15 | 3 | 3 | 1 | 1/3 | 3 | 0.25278 |
| | 15–20 | 4 | 3 | 3 | 1 | 3 | 0.41551 |
| | >20 | 3 | 3 | 1/3 | 1/3 | 1 | 0.16289 |

**Table 5.** *Cont.*

| Indicator | Class | Judgement Matrix | | | | | | | | Weight ($W_{ij}$) |
|---|---|---|---|---|---|---|---|---|---|---|
| Slope aspect | North | 1 | 1 | 1/5 | 1/7 | 1/8 | 1/9 | 1/3 | 1/2 | 0.02632 |
| | Northeast | 1 | 1 | 1/3 | 1/5 | 1/7 | 1/6 | 1/2 | 1 | 0.03591 |
| | East | 5 | 3 | 1 | 1/3 | 1/4 | 1/4 | 2 | 2 | 0.09014 |
| | Southeast | 7 | 5 | 3 | 1 | 1/2 | 2 | 4 | 4 | 0.22180 |
| | South | 8 | 7 | 4 | 2 | 1 | 3 | 5 | 7 | 0.33640 |
| | Southwest | 9 | 6 | 4 | 1/2 | 1/3 | 1 | 2 | 3 | 0.17164 |
| | West | 3 | 2 | 1/2 | 1/4 | 1/5 | 1/2 | 1 | 4 | 0.07541 |
| | Northwest | 2 | 1 | 1/2 | 1/4 | 1/7 | 1/3 | 1/4 | 1 | 0.04237 |
| Vegetation type | Hemerophyte | 1 | 4 | 1/2 | 5 | | | | | 0.37499 |
| | Broadleaf forest | 1/4 | 1 | 1/2 | 1 | | | | | 0.12538 |
| | Meadow | 2 | 2 | 1 | 3 | | | | | 0.39248 |
| | Mixed forest | 1/5 | 1 | 1/3 | 1 | | | | | 0.10715 |
| NDVI | 0–0.3 | 1 | 1 | 1/3 | 1/3 | 1/4 | | | | 0.08391 |
| | 0.55–0.65 | 1 | 1 | 1/3 | 1/2 | 2 | | | | 0.13792 |
| | 0.3–0.55 | 3 | 3 | 1 | 1 | 4 | | | | 0.35182 |
| | 0.65–0.75 | 3 | 2 | 1 | 1 | 3 | | | | 0.30628 |
| | 0.75–1 | 4 | 1/2 | 1/4 | 1/3 | 1 | | | | 0.12007 |
| Lithology | Q | 1 | 1/2 | 4 | 1 | 2 | 3 | 1/5 | | 0.12901 |
| | γ | 2 | 1 | 7 | 1 | 3 | 4 | 1/2 | | 0.21439 |
| | K + J | 1/4 | 1/7 | 1 | 1/3 | 1/2 | 3 | 1/5 | | 0.05090 |
| | Ar | 1 | 1 | 3 | 1 | 2 | 4 | 1/4 | | 0.14705 |
| | Q + Z | 1/2 | 1/3 | 2 | 1/2 | 1 | 2 | 1/3 | | 0.08317 |
| | β | 1/3 | 1/4 | 1/3 | 1/4 | 1/2 | 1 | 1/3 | | 0.04333 |
| | ∈ + O + Z | 5 | 2 | 5 | 4 | 3 | 3 | 1 | | 0.33215 |
| Distance from fault | 0–500 | 1 | 3 | 1/2 | 1/2 | 2 | | | | 0.19789 |
| | 500–1000 | 1/3 | 1 | 1/3 | 1/2 | 1/2 | | | | 0.08911 |
| | 1000–2000 | 2 | 3 | 1 | 2 | 2 | | | | 0.34454 |
| | 2000–3000 | 2 | 2 | 1/2 | 1 | 1 | | | | 0.20961 |
| | >3000 | 1/2 | 2 | 1/2 | 1 | 1 | | | | 0.15885 |
| Landform type | Fluvial terrace | 1 | 1/3 | 1/5 | 1/4 | 1/2 | 3 | 4 | | 0.08124 |
| | Undulating terrace | 3 | 1 | 1/2 | 1 | 2 | 4 | 3 | | 0.18835 |
| | Denudation of eroded hill | 5 | 2 | 1 | 3 | 5 | 4 | 4 | | 0.34318 |
| | Tectonic low hill | 4 | 1 | 1/3 | 1 | 3 | 2 | 5 | | 0.19120 |
| | Tectonic moderate hill | 2 | 1/2 | 1/5 | 1/3 | 1 | 3 | 2 | | 0.09903 |
| | Lava low terrace | 1/3 | 1/4 | 1/4 | 1/2 | 1/3 | 1 | 1 | | 0.05027 |
| | Lava plateau | 1/4 | 1/3 | 1/4 | 1/5 | 1/2 | 1 | 1 | | 0.04673 |
| Multi-year average precipitation | <720 | 1 | 3 | 4 | 3 | 4 | | | | 0.44926 |
| | 720–730 | 1/3 | 1 | 2 | 1 | 1/2 | | | | 0.13347 |
| | 730–740 | 1/4 | 1/2 | 1 | 1/2 | 1/3 | | | | 0.07666 |
| | 740–750 | 1/3 | 1 | 2 | 1 | 1/2 | | | | 0.13347 |
| | >750 | 1/4 | 2 | 3 | 2 | 1 | | | | 0.20713 |
| Distance from river | 0–100 | 1 | 3 | 2 | 2 | 4 | | | | 0.37497 |
| | 100–300 | 1/3 | 1 | 1/2 | 1/2 | 2 | | | | 0.12081 |
| | 300–600 | 1/2 | 2 | 1 | 1 | 3 | | | | 0.21536 |
| | 600–1000 | 1/2 | 2 | 1 | 1 | 3 | | | | 0.21536 |
| | >1000 | 1/4 | 1/2 | 1/3 | 1/3 | 1 | | | | 0.07350 |
| Distance from road | 0–100 | 1 | 3 | 2 | 1/2 | 4 | | | | 0.28286 |
| | 100–300 | 1/3 | 1 | 1/2 | 1/4 | 2 | | | | 0.10469 |
| | 300–600 | 1/2 | 2 | 1 | 1 | 3 | | | | 0.21437 |
| | 600–1200 | 2 | 4 | 1 | 1 | 3 | | | | 0.32492 |
| | >1200 | 1/4 | 1/2 | 1/3 | 1/3 | 1 | | | | 0.07316 |
| Mining point density | 0–5 | 1 | 3 | 2 | 3 | 0.5 | | | | 0.27758 |
| | 5–9 | 1/3 | 1 | 1/2 | 1/2 | 1/2 | | | | 0.09473 |
| | 9–13 | 1/2 | 2 | 1 | 1/3 | 1/2 | | | | 0.12500 |
| | 13–21 | 1/3 | 2 | 3 | 1 | 1/3 | | | | 0.16494 |
| | 21–31 | 2 | 2 | 2 | 3 | 1 | | | | 0.33774 |

### 3.1.3. Results of the AHP-ICM Model

Based on the results of the actual survey in the study area, a combination of subjective and objective methods was used to determine the hazard of collapse in Huinan County, taking into account the geographic and geological environment and external factors that contribute to the disaster. The comprehensive information content (Table 6) amount of each sub-class in each hazard evaluation indicator was calculated using Equation (9), and then the sub-class of each evaluation indicator was assigned a value using a raster calculator,

finally superimposed, and the natural breaks classification method was used to divide it into five hazard classes: very low, low, medium, high, and very high, to obtain a hazard zoning map, and the results are shown in Figure 10c. As can be seen from Table 6, the CIs for multi-year average precipitation (730–743 mm), distance from road (0–100 m), and lithology (γ) are the largest at 0.29487, 0.22677, and 0.17298, respectively, which causes the very high hazard zones to be mainly located in the central part, in the northwestern part, along the northeastern road, and in the west-southwestern part of the study area, which makes the collapse disaster prone to occur in this part of the area. The main reason is that these areas are located in the middle and low mountainous belt and are affected by the long-term frequent economic activities of human beings. According to the statistics of the number of disaster points, it is concluded by comparison that the collapse disasters in Huinan County are mainly distributed in the tectonic low hill geomorphic area and hilly area, and the lithology is mostly limestone, granite, and metamorphic rock groups, which have low vegetation coverage, and the human activities are the strongest in these areas. In tectonic mesas and high terraces, human activity is less, elevation is greater, vegetation cover is greater, and collapse disasters are least developed.

**Table 6.** Combined information content for each collapse evaluation indicator's sub-classes.

| Indicator | Class | CI | Indicator | Class | CI |
|---|---|---|---|---|---|
| Slope angle | 0–5 | −0.00980 | Distance from river | 0–100 | 0.12236 |
|  | 5–10 | −0.00641 |  | 100–300 | 0.01706 |
|  | 10–15 | 0.01256 |  | 300–600 | 0.05029 |
|  | 15–20 | 0.00744 |  | 600–1000 | 0.02330 |
|  | >20 | −0.02392 |  | >1000 | −0.18255 |
| Slope aspect | North | 0.00000 | Distance from road | 0–100 | −0.31904 |
|  | Northeast | −0.00654 |  | 100–300 | −0.01884 |
|  | East | −0.00664 |  | 300–600 | −0.02012 |
|  | Southeast | 0.00145 |  | 600–1200 | −0.02608 |
|  | South | 0.00011 |  | >1200 | 0.22677 |
|  | Southwest | 0.00371 | Vegetation type | Hemerophyte | 0.00764 |
|  | West | −0.00534 |  | Broadleaf forest | −0.00073 |
|  | Northwest | −0.00628 |  | Meadow | 0.00464 |
| Multi-year average precipitation | <720 | −0.24059 |  | Mixed forest | −0.00726 |
|  | 720–730 | 0.12476 | NDVI | 0–0.3 | 0.03751 |
|  | 730–740 | 0.29487 |  | 0.3–0.55 | 0.02307 |
|  | 740–750 | −0.04157 |  | 0.55–0.65 | −0.01345 |
|  | >750 | −0.41161 |  | 0.65–0.75 | −0.02982 |
| Lithology | Q | −0.09747 |  | 0.75–1 | −0.06986 |
|  | γ | 0.17298 | Mining point density | 0–5 | −0.02159 |
|  | K + J | 0.00983 |  | 5–9 | −0.00041 |
|  | Ar | −0.08135 |  | 9–13 | 0.02131 |
|  | Q + Z | −0.01542 |  | 13–21 | 0.02999 |
|  | β | −0.00926 |  | 21–31 | 0.01626 |
|  | ∈ + O + Z | 0.14407 | Distance from fault Distance from fault | 0–500 | 0.00991 |
| Landform type | Fluvial terrace | −0.01292 |  | 500–1000 | 0.01350 |
|  | Undulating terrace | −0.01505 |  | 1000–2000 | −0.01114 |
|  | Denudation of eroded hill | 0.02344 |  | 2000–3000 | 0.01520 |
|  | Tectonic low hill | 0.03099 |  | >3000 | −0.02904 |
|  | Tectonic moderate hill | −0.04047 |  | 0–500 | 0.00991 |
|  | Lava low terrace | −0.00976 |  |  |  |
|  | Lava plateau | −0.02913 |  |  |  |

### 3.2. Validation of the Hazard Mapping Model

The results obtained from the ICM, AHP, and AHP-ICM models were validated to guarantee the accuracy of the collapse hazard mapping model, and then the optimal model was selected for the study area collapse risk mapping. In this study, the three models were validated by applying the receiver operating characteristic (ROC) curve in Origin 2022 software. The curve is a two-dimensional plot and is widely used in the validation of two-dimensional classification models [18,19,25], and the evaluation process is simple and easy to operate. The ROC uses the area under the curve (AUC) to directly determine the accuracy of the model. The results are intuitive, with AUC values between 0.5 and 1; the closer the result is to 1, the more accurate the model and the better the performance. In this study, ROC curves were plotted based on the principle of ROC curves with an equal number of randomly selected collapse and non-collapse points. The ROC results in Figure 11 show that the AHP-ICM model has the highest accuracy (AUC = 87.4%), followed by the ICM model (AUC = 85.6%) and the AHP (AUC = 80%).

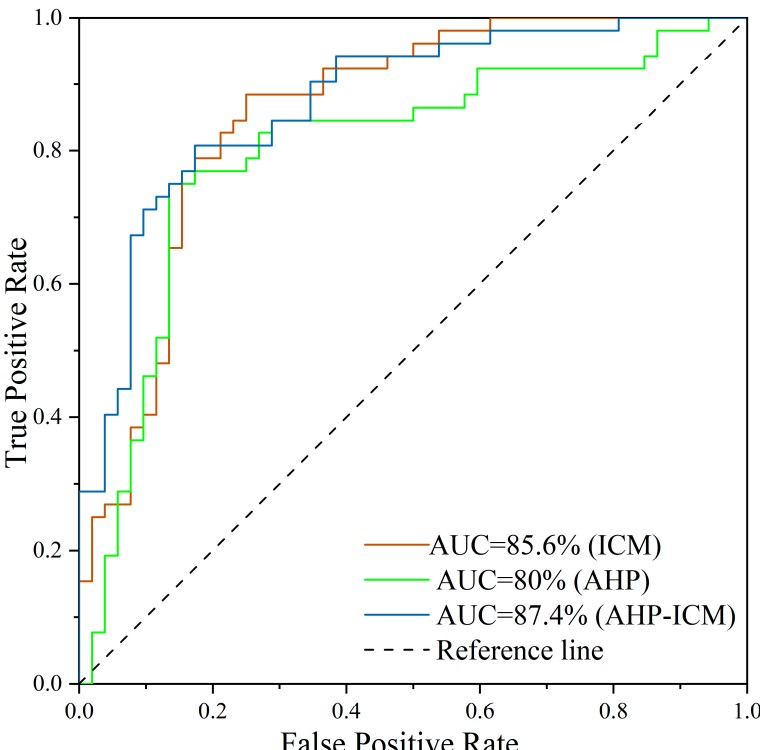

**Figure 11.** Receiver operating characteristic (ROC) curve.

### 3.3. Comparison of Hazard Mapping Models

Three models were used to compare the hazard mapping in this study: the ICM model, the AHP model, and the AHP-ICM model. The accuracy of the three models was verified by ROC curves, and the results in Figure 11 show that the AHP-ICM model has the highest accuracy. In addition, the results of the mapping of the three models are compared in more detail according to the five hazard classes, and the statistical results are shown in Figure 12.

The collapse hazard map produced by the natural breaks classification method should meet the following requirements: (1) The areas classified as high and very high hazard should cover as small an area of the study area as possible. (2) The areas with high and very high hazards should contain as many known collapse points as possible. The results show that the AHP model has the smallest total proportion of very high and high hazard classes (28.03%), but it contains the fewest known collapse points (37). It is followed by the AHP-ICM model with a total proportion of 28.88% of very high and high classes and 44 known collapse points, and then the ICM model with a total proportion of 32.51% of very

high and high classes and 48 known collapse points. The last model is the ICM model, with a total of 32.51% of very high and high grades and 48 known collapse points. Although the ICM model contains the highest number of collapse points in areas with very high and high hazard classes, their total percentage is the highest of the three models, while the AHP-ICM model contains seven more known collapse points than the AHP model with a total percentage of very high and high hazard classes of only 0.85% more than the AHP model. In terms of disaster point density, the highest density in areas with very high and high hazard classes is found in the AHP-ICM model (6.69 per 100 km²), followed by the ICM model (6.48 per 100 km²) and the AHP model (5.8 per 100 km²).

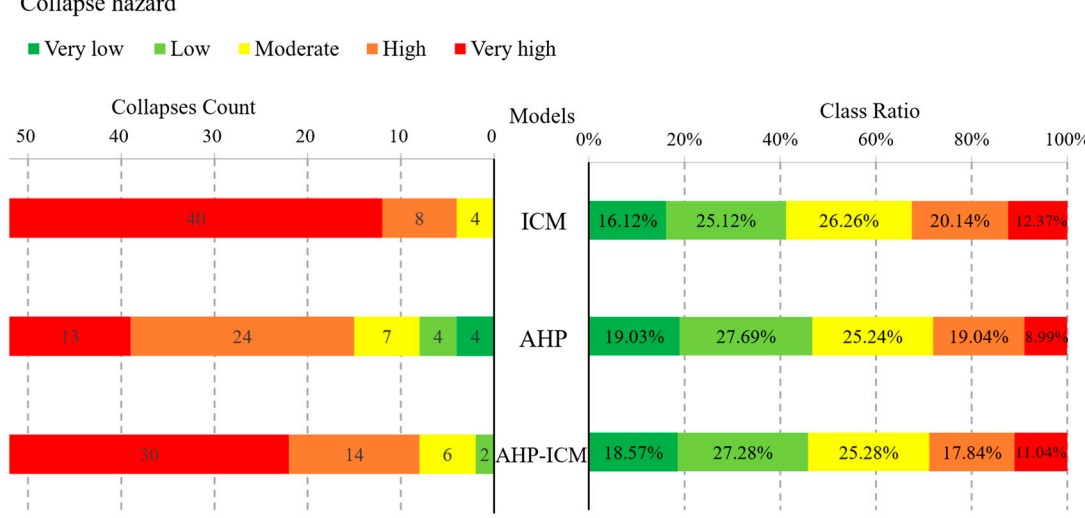

**Figure 12.** Statistics of the results of the collapse hazard mapping.

By comprehensively analysing the ROC curves, the percentage of high and very high hazard classes of the models, and the disaster point density, this study determines the degree of superiority of the three hazard mapping models, of which the optimal model is the AHP-ICM model, which is used in the study of collapse risk mapping (see Section 5).

## 4. Results of Exposure, Vulnerability, and Emergency Response and Recovery Capability Mapping

This study also used a combination of subjective and objective methods to analyse the exposure, vulnerability, and emergency response and recovery capability mappings. Weighting of the 10 evaluation indicators in the exposure, vulnerability, and emergency response and recovery capability indicator evaluation system was performed using EWM using Excel 2021 software; then, using AHP, judgment matrices were constructed to calculate the subjective weights of each of the five sub-classes in the evaluation indicators, and a total of 10 judgment matrices were constructed to derive the weight values of each sub-class; finally, the weights ($We$) of the evaluation indicators are coupled with the sub-class weights ($W_{ij}$) in the evaluation indicators using Formula (15) to obtain the sub-class comprehensive weights $S$. The results of the overlay are shown in Figure 13. The results show that the high and very high exposure areas are mostly concentrated in densely populated and built-up areas, the reason being that these areas have increased the scale of engineering and construction activities due to economic development and farming and reclamation, increasing the potential for collapse disasters to occur and threaten nearby areas. High and very high vulnerability areas are mainly located in rural and township areas close to the mountains, the reason being that the vulnerable population and dwellings are more numerous in this area and are most vulnerable to collapse disasters, and they are less likely to recognise the approaching danger or move slowly when a collapse disaster occurs. The

closer to urban and rural areas, the more relief agencies are available, and the more money is invested in disaster management, the better their ability to prevent and mitigate disasters.

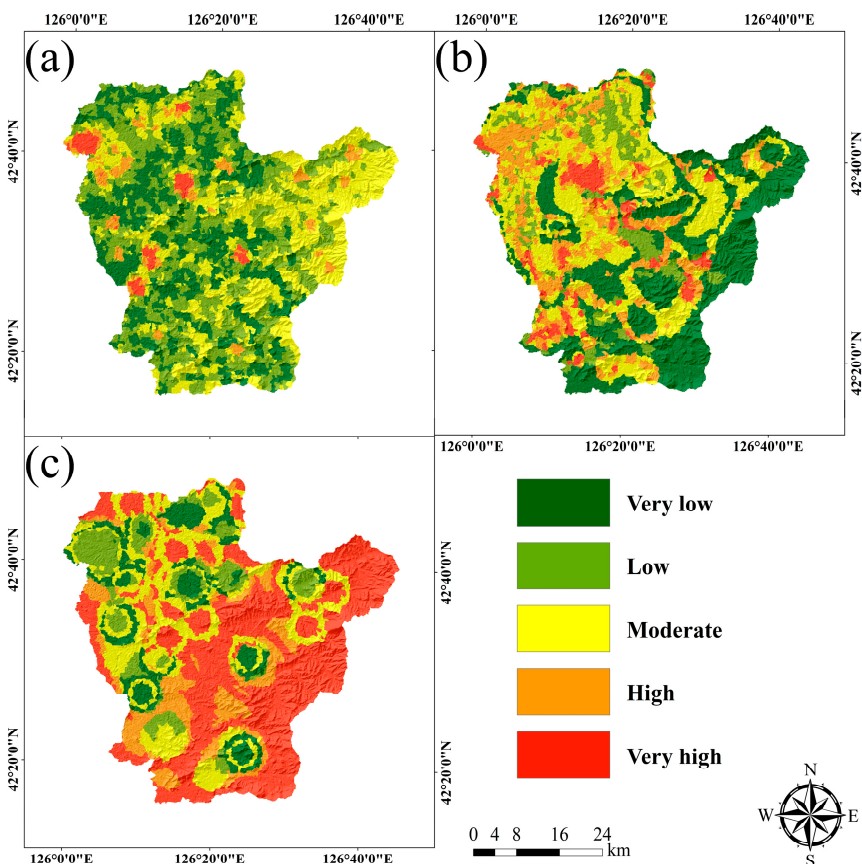

**Figure 13.** Results of the study area mapping based on the AHP-EWM model: (**a**) exposure overlay; (**b**) vulnerability overlay; and (**c**) emergency response and recovery capability overlay.

## 5. Results of Risk Mapping

Based on the above analysis, the optimal model for collapse hazard mapping is the AHP-ICM model. The final four elements of overlay maps are as follows: hazard mapping based on the AHP-ICM model, exposure overlay, vulnerability overlay, and emergency response and recovery capability mapping based on the AHP-EWM model.

Combining the four elements obtained by overlay, the collapse disaster risk index was calculated according to Equation (16) and classified using the natural breaks classification method into four classes: very high, high, moderate, and low, to obtain a collapse disaster risk zoning map, as shown in Figure 14. Of these, the very high-risk area accounts for 6.06% and contains 20 collapse points, mainly in the central, northwestern, southwestern, and northeastern parts of the study area along the roads. The very high-risk areas are mainly caused by the construction of the road, which causes many cut slopes, and by the mining and cultivation of the land, which destroys a lot of vegetation. The high-risk zone covers 30.07% of the area and contains 28 landslides, which are caused by the low topography and low vegetation cover in these areas and by the long-term rainfall and the incompatibility of human engineering construction with the geological environment. The low- and medium-risk areas, which together account for 63.87% of the study area, contain four collapse points and are mainly located in areas of high elevation and low elevation areas where human engineering activities are less frequent.

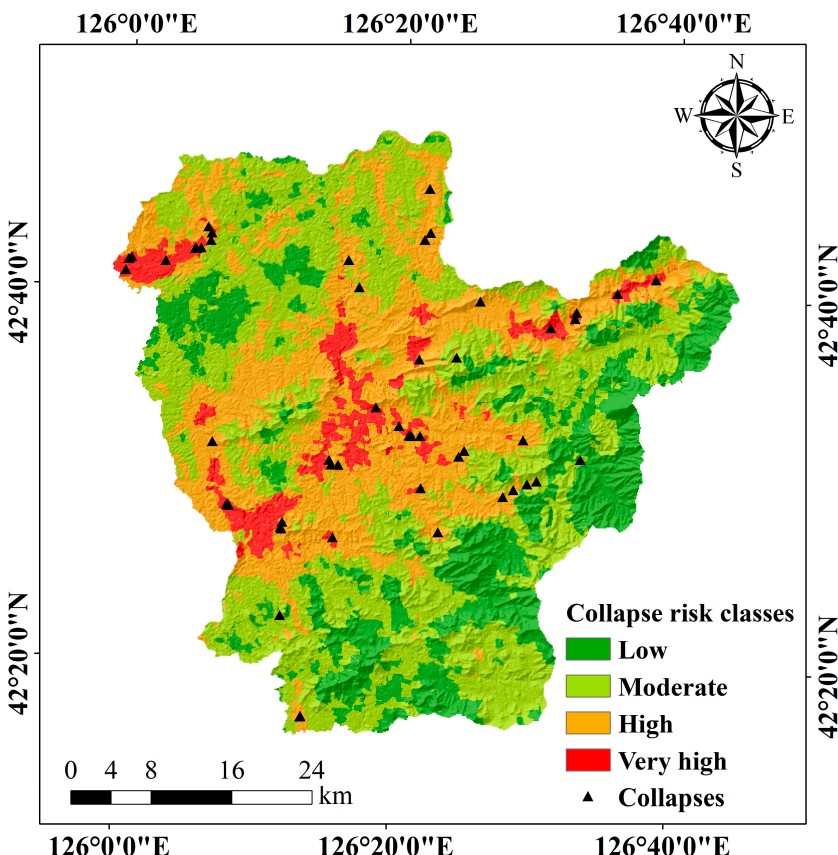

**Figure 14.** Collapse disaster risk zoning map.

## 6. Discussion

### 6.1. Importance and Significance of This Study

Huinan County is located in a semi-mountainous area, with large changes in topography and geomorphology, various types of rock and soil bodies, and development of local geological formations, superimposed on the role of strong human engineering activities such as road building, urban construction, and mineral resource extraction, geological disasters are very well developed, especially under the influence of extreme abnormal climate, which triggers or aggravates collapse disasters, so disaster prevention and control are still facing great pressure. Therefore, risk assessment and mapping of collapse disasters are of great practical significance. In this study, based on the formation principle of natural disaster risk, the hazard mapping model of collapse disaster was established using the ICM, AHP, and AHP-ICM, and the mapping model of exposure, vulnerability, and emergency response and recovery capability was established using AHP-EWM. The risk mapping model combines multiple evaluation indicators and comprehensively considers multiple dimensions of collapse risk, which is scientific and operable. This study provides a comprehensive assessment and accurate identification of the risk of collapses in Huinan County, which provides an important reference for collapse disaster management and decision making and has far-reaching practical significance for the prevention and mitigation of geological disasters in mountainous areas. In addition, the methods and models used in this study have some value and potential for application in risk assessment and mapping studies in other regions. It is worth mentioning that previous studies have mainly focused on the evaluation of disaster hazard, exposure, and vulnerability, while few studies have been conducted on the evaluation of emergency response and recovery capability, and there are few sources of data and ways to visualise their data. In this study, three socio-economic indicators, namely education investment, local financial revenue, and relief agencies' ca-

pacity, are used as indicators of emergency response and recovery capability, which play a key role in collapse risk mapping.

### 6.2. Comprehensive Evaluation of Hazard Mapping Model

The ICM method is a relatively objective evaluation model that establishes a suitable evaluation indicator system and its sub-class through the actual survey results of the study area and then calculates the information value of each indicator's sub-class, which reduces the influence of subjectivity [24,44] but ignores the relationship between the sub-class of evaluation indicators and the occurrence of collapse. The AHP method is a subjective weighting method [20,54], and the assignment of its weights requires sufficient experience and level. Because the established hierarchy and calculation of weights are more complicated, it is easy to calculate improper weights in a certain sub-class, which is difficult to avoid, so this leads to a lower accuracy of collapse hazard mapping. The AHP-ICM method, as a combined subjective and objective method [18,19,50], compared with a single method, can ensure both the risk mapping objective and scientific soundness but also allows for subjective analysis based on the actual situation in the study area. There have been researchers using the AHP-ICM method in the landslide sensitivity mapping study. Wang et al. [18] used the AHP-ICM method to carry out sensitivity analyses of 41 landslides, and more accurate results were obtained, which verified the correctness and reasonableness of the method. Du et al. [19] used the AHP-ICM method to analyse the landslide sensitivity mapping of 799 landslides in the Eastern Himalayan zone of Tibet, and the results showed that the AHP-ICM method has a high prediction accuracy. Ma et al. [24] compared the AHP-ICM method with other combined methods in landslide sensitivity mapping and found that the prediction accuracy of the AHP-ICM method was similar to the other methods. In the same field, although the AHP-ICM method still has a certain gap compared with existing machine learning methods (Support Vector Machine, Random Forest, Artificial Neural Networks, etc.), the AHP-ICM method is much easier to compute and operate and only needs to extract the relevant landslide data of the evaluation indicators in the GIS, and then the corresponding weights of the indicators can be derived by calculating them directly in the Excel 2021. Machine learning methods, on the other hand, are more demanding in terms of data quality and feature selection, requiring adequate data preparation and model training processes, while the models are less explanatory, making it difficult to explain the key features identified by the models.

### 6.3. Limitations and Perspectives of This Study

This study considers as comprehensively as possible the evaluation indicators related to collapse disasters from a four-element perspective; however, the data used are based on existing datasets and may be incomplete and inaccurate for economic and education-related data. In future work, more high-quality data can be further collected to improve the accuracy of the findings. In future collapse risk mapping, more indicators related to collapse risk should be considered, and earthquakes have a certain impact on it. There has been no seismic activity in this study area in 40 years, and seismic data are difficult to obtain, so the impact of earthquakes is not considered in this paper. In areas with high seismic impacts, it is important to incorporate seismic factors into collapse risk mapping in order to better predict and mitigate the occurrence of collapse disasters. For the impact of indicators on collapse, the distance category should be further improved based on the survey results to narrow the buffer zone and improve the accuracy of collapse risk zoning. For areas at high risk of collapse, the prevention, monitoring, and emergency rescue of collapse disasters should be strengthened, and the potential risk of collapse disasters should be reduced by means of land planning, engineering measures, and warning and education.

## 7. Conclusions

In this paper, the curvature watershed method is used to make mapping units, which are used to extract all evaluation indicators. Then, the ICM, AHP, and AHP-ICM models

are applied to hazard mapping, the accuracy of the three models is verified by the ROC curve, the optimal model is selected, the zoning of exposure, vulnerability, and emergency response and recovery capability made by applying the AHP-EWM model is combined, the collapse disaster risk index model used in the risk index is calculated, and finally, the collapse risk zoning map was drawn using GIS. This led to the following conclusions.

Based on the evaluation indicator system determined by the four elements of natural disaster risk theory, TOL and VIF were used to analyse the covariance among the indicators, and it was verified that there was independence among all 21 indicators, indicating that the established evaluation indicator system was reasonable.

The ICM model, AHP model, and AHP-ICM model were compared in the collapse hazard mapping, and they were validated by the ROC curve, yielding AUC values for the three models: ICM model (85.6%), AHP model (80%), and AHP-ICM model (87.4%). After a comprehensive comparison, the percentage of the ICM, AHP, and AHP-ICM models with very high and high hazard ratings are 28.03%, 28.88%, and 32.51%, respectively, and the number of known collapses they contain are 48, 37, and 44, respectively, with disaster point densities of 4.48, 5.8, and 6.69 per km$^2$, respectively; the results show that the AHP-ICM model is the optimal model for collapse hazard mapping in Huinan County.

The AHP-EWM model was used to map the exposure, vulnerability, and emergency response and recovery capability of the collapse disaster bearers. The results show that very high exposure and high exposure areas are mostly concentrated in densely populated and built-up areas, very high vulnerability and high vulnerability areas are mainly located in rural and township areas close to the mountain, and areas with high emergency response and recovery capability are mostly concentrated in urban and township areas.

According to the results of the collapse risk mapping, the very high-risk zone accounts for 6.06% of the study area, containing 20 collapse points and a disaster point density of 14.49 per 100 km$^2$; the high-risk zone covers 30.07% of the area, containing 28 collapse points and a disaster point density of 4.09 per 100 km$^2$. Very high-risk areas and high-risk areas are mainly located in (1) Northwestern Huinan County (Chaoyang Township), (2) the central part (Huinan Town–Sansonggang Town–Fumin Town), (3) the southwest (look-alike whistle town), and (4) the northeast (Shidaohe Town). When preventing and controlling collapse disaster sites or sections, decision makers should consider these four areas first in order to reduce disaster losses and casualties.

**Author Contributions:** Conceptualisation, Zengkang Lu; data curation, Zengkang Lu and Jie Wang; formal analysis, Zengkang Lu and Chenglong Yu; methodology, Zengkang Lu; writing—original draft, Zengkang Lu; funding acquisition, Yichen Zhang and Huanan Liu; writing—review and editing, Chenglong Yu, Yanan Chen and Jiquan Zhang. All authors have read and agreed to the published version of the manuscript.

**Funding:** This research was funded by the Key Scientific and Technology Research and Development Program of Jilin Province (grant No. 20220203002SF), Jilin Science and Technology Program (grant No. 20230203130SF), Jilin Education Program (grant No. JJKH20230722KJ).

**Data Availability Statement:** The codes and data for this article are freely available at https://www.gscloud.cn/ (accessed on 30 February 2023), https://www.webmap.cn/ (accessed on 9 March 2023), https://www.resdc.cn/ (accessed on 15 March 2023), https://www.databox.store (accessed on 11 February 2023), https://www.worldpop.org/ (accessed on 14 February 2023), and http://www.geodata.cn/ (accessed on 16 March 2023).

**Conflicts of Interest:** The authors declare no conflict of interest.

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
