# Peer review of "Application of AHP-ICM and AHP-EWM in Collapse Disaster Risk Mapping in Huinan County"

_ijgi, doi:10.3390/ijgi12100395_

Round 1
Reviewer 1 Report
Comments and Suggestions for Authors
The manuscript entitled “Application of AHP-ICM and AHP-EWM in Collapse Disaster Risk Mapping in Huinan County” presents an approach to estimate the risk using various Geospatial approaches including AHP, ICM, EWM etc. However, the manuscript needs to be improved for possible publication. The structure of the manuscript's flow could be improved. The presentation of data collection methods lacks conciseness and clarity, making it difficult for readers to understand the specific sources and processing techniques used
The delineation of slope units in the "Mapping Unit" section could be more coherent and logically structured, providing a clearer understanding of the methodology.
The methodology lacks detailed explanations of the formulas and underlying assumptions of the models used. Authors mentioned the ICM, AHP, and AHP-ICM models, however, they fail to provide sufficient information on the specific calculations and formulas used within these models. This lack of detail compromises the scientific rigor of the study and makes it difficult for readers to assess the validity and reliability of the findings.
The result session need more attention. In the "Results of the Hazard Mapping Model" section, the authors provide a thorough description of the ICM and AHP models, but the AHP-ICM model, identified as the optimal model, is not given the same level of detail. A comparative analysis of the three models and a clear explanation of their respective strengths and weaknesses would have enhanced the presentation of the findings.
The final discussion also falls short in several aspects. While the conclusion summarizes the study's objectives and major findings, it lacks a comprehensive and insightful discussion of the practical implications of the results. Recommendations for collapse risk mitigation strategies and discussions on the limitations of the study are notably absent. Moreover, suggestions for further research avenues or directions are lacking, which could have provided valuable insights for future studies in this field.
Reviewer 2 Report
Comments and Suggestions for Authors
Thank you for this interesting article. Please modify the figure 9, it is difficult to read because the letters are very small.
Reviewer 3 Report
Comments and Suggestions for Authors
The reviewed article is extremely interesting and valuable in terms of knowledge and methodology. The authors have included beautifully drawn figures, the captions under them are exhaustive. The article ends with properly formulated conclusions. The text of the article should be published in the nearest time.
